# Implicit Neural Representation Inference for Low-Dimensional Bayesian Deep Learning

**Panagiotis Dimitrakopoulos**[1]**, Giorgos Sfikas**[2] **& Christophoros Nikou**[1]

[1]Dept. of Computer Science & Engineering, University of Ioannina, Ioannina Greece
[2]Dept. of Surveying & Geoinformatics, University of West Attica, Athens Greece
[1]{p.dimitrakopoulos,cnikou}@uoi.gr,[2]gsfikas@uniwa.gr

## Abstract

Bayesian inference is the standard for providing full predictive distributions with well calibrated uncertainty estimates. However, scaling to a modern, overparameterized deep learning setting typically comes at the cost of severe and restrictive approximations, sacrificing model predictive strength. With our approach, we factor model parameters as a function of deterministic and probabilistic components; the model is solved by combining maximum a posteriori estimation of the former, with inference over a low-dimensional, Implicit Neural Representation of the latter. This results in a solution that combines both predictive accuracy and good calibration, as it entails inducing stochasticity over the full set of model weights while being comparatively cheap to compute. Experimentally, our approach compares favorably to the state of the art, including much more expensive methods as well as less expressive posterior approximations over full network parameters.

## 1 Introduction

Bayesian Neural Networks (BNNs) are a class of models that propose elegant solutions to the pathologies of standard NNs (Ritter et al., 2018a; Jospin et al., 2022; Gawlikowski et al., 2023). In BNNs, model parameters are defined as random variables that follow a prior (posterior) distribution, which encodes knowledge about the model before (after) having "seen" the training data. Learning is cast as an inference problem, where the task is to compute efficiently the posterior distribution. In turn, making predictions on new data is replaced by computing a predictive distribution. Advantages include that uncertainty estimates are calibrated and robust, and hyperparameter estimation can be performed through a principled evidence maximization framework. In BNNs, Bayesian inference is not exact, and a direct application of Bayes' law leads to an intractable computation. An approximation has to be applied, and in this respect numerous solutions have been proposed. A factor that complicates this problem is that the approximation must lead to a scalable, practical implementation that must take into account that the data and model size may be far larger than what was the norm in methods and models that dominated Bayesian inference in the pre-deep learning era. Several solutions have been proposed in this respect, rehashing and adapting older methods (Betancourt, 2017; Daxberger et al., 2021a) or putting forward completely fresh approaches (Maddox et al., 2019).

Implicit Neural Representations (INRs) are related to a different line of research that is orthogonal to that involving Bayesian networks (Sitzmann et al., 2020; Dupont et al., 2021b). With INRs, the goal is to represent a signal in terms of a trained neural network. Unlike standard representations as discrete sets of values over a canonical grid, an INR accepts continuous coordinates as inputs. Therefore, the INRs allow for a continuous representation, with the underlying NN providing values of the represented signal at theoretically any granularity. Related breakthroughs in improving representation of high frequencies have contributed to the popularity of the approach (Sitzmann et al., 2020; Mildenhall et al., 2021). Numerous signal representation use-cases have been explored, including images, video, 3D shapes or Neural Radiance Fields (NeRFs). With the latter, a NN is tasked with mapping ray position and direction to color and density values. Part of the parameters of a larger NN can also be encoded with an INR; in Romero et al. (2021a), convolutional kernels are represented in terms of Multiplicative Anisotropic Gabor Networks. In this case, the implicit representation allows for kernels that generalize well to higher resolutions than the ones originally trained with. Aside from allowing for continuous representation at multiple scales, another major focus involves the

INR's capability of producing a compressed, low-dimensional representation (Benbarka et al., 2022; Strümpler et al., 2022).

In this work, we propose a class of Bayesian Neural Network that is parameterized using a combination of deterministic and stochastic parameters. In recent work, similar partitions are employed (Kristiadi et al., 2020; Dusenberry et al., 2020; Daxberger et al., 2021b), where a specific subnetwork is set to be stochastic while the rest of the network is deterministic. Unlike these works, we define *all* parameters as functions conditioned over both deterministic and probabilistic components. Normally, this is very much desired but computationally prohibitive due to the huge number of parameters in modern NNs; in our work, this is made feasible due to the probabilistic component being parameterized through an INR hypernetwork, which compresses probabilistic factors through a low-dimensional SIREN representation (Sitzmann et al., 2020). It is over this representation that we assume a prior distribution, and perform inference. As the number of probabilistic factors is kept low, we are allowed to make fewer concessions w.r.t. constraining the form of the posterior and the predictive. The result is a process that is comparatively closer to exact inference, leading to more accurate estimates and better uncertainty calibration.

In a nutshell, the deterministic model component is responsible for ensuring accurate results, while the low-dimensional probabilisic component is responsible for inducing stochasticity to the entirety of the network. We validate our claims and model across a variety of experimental trials, where we show that our model produces accurate and well-calibrated uncertainty estimates.

## 2    BACKGROUND AND MOTIVATION

We consider a supervised learning setting, where we have a training dataset $D = \{X, Y\}$, with inputs $X = \{x_n\}_{n=1}^N$ and outputs $Y = \{y_n\}_{n=1}^N$, and we define a mapping $g_w : \mathcal{X} \to \mathcal{Y}$, where $\mathcal{X}$ and $\mathcal{Y}$ are the input and output domains respectively. This mapping is modelled as a NN with parameters (weights and biases) $w \in \mathbb{R}^{d_w}$. Under the BNN paradigm, we assume that the mapping parameters are probabilistic, so we can say that they follow some prior distribution $p(w)$, while we aim to compute (in practice, estimate) their posterior distribution $p(w|D)$. Given the posterior distribution, we can then opt to find the predictive distribution for some unseen datum $x^\star$, formally:

$$p(y|D, x^\star) = \int p(y|g_w(x^\star))p(w|D)dw. \tag{1}$$

In contrast to a Maximum a Posteriori (MAP) solution, which would optimize a cost combining log-likelihood and log-prior terms:

$$\bar{w} = \arg\max_w [\log p(y|g_w(x)) + \log p(w)], \tag{2}$$

a Bayesian solution aims to compute distributions for both the posterior and the predictive.

Several options are available to proceed. The Stochastic Weight Averaging-Gaussian method (SWAG) assumes a Gaussian posterior for the weights, with the distribution mean and covariance approximated as a function of the objective optimization method (Stochastic Gradient Descent) with a modified learning schedule (Maddox et al., 2019). Laplace Approximation (LA) also assumes Gaussian distributed parameters, with a precision matrix that is computed as the negative Hessian of the loss. After having a weight posterior, an option can be to sample the predictive distribution and either obtain point estimates for test data, or perform Bayesian model averaging (Maddox et al., 2019). Additional simplifying assumptions can lead to a closed form also for the predictive. The Generalized Gauss-Newton approximation is closely related to a linearizing assumption for the output layer of the NN (Immer et al., 2021), which conveniently leads to a Gaussian approximation for the predictive distribution. The covariance of the predictive is then dependent on a combination of two factors: the covariance of the posterior (negative loss Hessian in LA) and the Jacobian for the specific point. Interestingly, a relation between the subspace spanned by the SGD trajectory vectors (used by SWAG) and the corresponding one to the most important eigenvectors of the Hessian (used by LA) is discussed in Gur-Ari et al. (2018). Normalizing Flows (NFs) represent a powerful framework for density estimation (Dinh et al., 2016), that may in principle also be used to model the posterior of a large NN.

Scalability is a crucial factor when it comes to learning methods in the context of NNs. Assuming an entire network to be probabilistic implies significant overhead in terms of various factors. Common

remedies include assuming a Gaussian form combined with a low-rank approximation of the Hessian, and using a simplified, even diagonal covariance structure. Kronecker-Factored Approximate Curvature (KFAC) expresses a useful tradeoff, which neglects only cross-layer correlations and uses a block-diagonal covariance matrix (Ritter et al., 2018b). Another option involves treating only part of the network as non-deterministic (Kristiadi et al., 2020; Daxberger et al., 2021b). We then have uncertainty only in the last layer neurons, treating the rest of the network as a feature extractor (Weber et al., 2018). As a consequence, and to the degree that these assumptions are overly simplistic, the approximate distributions may turn out to be very far from the actual posterior and predictive. This often translates to a dramatic reduction of predictive strength in practice.

In the following Section, we shall discuss our approach to dealing with these issues.

## 3 PROPOSED MODEL: LOW-DIMENSIONAL BAYESIAN DEEP LEARNING

We propose to move from the high-dimensional setting of full inference in a modern Neural Network to low-dimensional inference, by assuming an auxiliary Implicit Neural Representation alongside the main network. We perform density estimation over the parameters of the INR hypernetwork, while treating the factors corresponding to the original weights as deterministic parameters. This allows us to employ powerful inference methods (we discuss LA, SWAG, NFs) with minimal approximation concessions, by leveraging on the small size and representational strength of the INR.

### 3.1 INR MODELING

Given the NN that models the mapping $g_w$ (cf. Section 2), the first step of our approach is to augment each weight $w$ with a multiplicative nuisance factor $\xi$ (Srivastava et al., 2014; Kingma et al., 2015; Louizos & Welling, 2017). In particular, we use $w \circ \xi$, where $\circ$ is point-wise multiplication, and the dimensionality of $\xi$ is identical to that of $w$.

The $\xi$ factor is parameterized using an INR (Dupont et al., 2022), obtained as the output of a function $f_{w_{INR}} : I \to \mathbb{R}$, where tensor coordinates (domain $I$) are mapped to layer values. More specifically for a convolutional main network, the INR hypernetwork learns a mapping from a 5 dimensional $I$ to a scalar value which corresponds to the nuisance factor associated with the weight $w_{c,o,k_i,k_j,l}$ located at the kernel position $k_i, k_j$ at channel $c$ of filter $o$ in layer $l$ of the main/primary network (in the case of a fully-connected layer, dimensions $k_i$ and $k_j$ are omitted). With the above modeling choice, the hypernetwork can be easily shared across each layer of the main network and reduce the overall modeling complexity. The architecture of the INR is defined as a multi-layer perceptron with sinusoidal activations, as with the SIREN model of Sitzmann et al. (2020). Formally, the input vector $z_{i-1}$ for layer $i$ is transformed according to $z_{i-1} \to sin(\omega_0(w^i_{INR}z_{i-1} + b^i_{INR}))$, where $w^i_{INR}, b^i_{INR}$ denote weights and biases of the INR layer $i$, and $\omega_0$ is a fixed hyperparameter.

In INRs, any target quantity can be modelled regardless of its size, while in traditional networks parameter size is coupled with target dimensionality. This characteristic, in combination with the stochastic character of $\xi$ allows us to choose the complexity of $f_{w_{INR}}(\cdot)$ to be (much) lower than that of its target ($d_{w_{INR}} \ll d_\xi$). Thus, $w_{INR}$ parameters can also be interpreted as a low-dimensional representation of factors $\xi$.

### 3.2 BAYESIAN INFERENCE

In our method, we treat the product $w \circ \xi$ as a stochastic random variable coming from a parametric distribution $p(w, \xi) = p(w)p(\xi)$. Here we are taking advantage of the INR hypernetwork modeling of $\xi$ and implicitly place a prior over those variables, by defining a prior over the INR parameters $w_{INR}$. This allows us to reason about $\xi$ but in the much lower dimensional space of $w_{INR}$. Following the supervised learning setting of Section 2, our aim remains to compute the posterior $p(w, w_{INR}|D)$. Since the posterior distribution cannot be obtained in closed form, we cannot apply exact inference methods. Thus we resort to approximate inference, under an additional assumption that we only require a deterministic estimate over $w$. We encode this constraint as a factorization over separate approximate posterior distributions $q(w)$ and $q(w_{INR})$, where $q(w) = \delta(w - \underline{w})$, and $\delta(\cdot)$ is the Dirac delta function. This forces $w$ to be deterministic, equal to a point estimate $\underline{w}$. The full approximate

posterior is then written as:

$$p(w, w_{INR}|D) \approx q(w, w_{INR}) = q(w_{INR})q(w) = q(w_{INR})\delta(w - \underline{w}). \tag{3}$$

**Laplace Approximation.** One way to proceed is by constructing a Laplace approximation over $q(w_{INR})$. We approximate $p(w_{INR}|D)$ by

$$q(w_{INR}) = \mathcal{N}(\bar{w}_{INR}, \Lambda^{-1}), \tag{4}$$

$$\Lambda = C^{-1} + \sum_{n=1}^{N} \nabla^2_{w_{INR}} \log p(y_n|g_{w,w_{INR}}(x_n))|_{\bar{w}_{INR}}, \tag{5}$$

where we have assumed a prior $w_{INR} \sim \mathcal{N}(0, C)$. Mean $\bar{w}_{INR}$ is found as the Maximum a Posteriori solution (eq. 2). Under this scheme, $q(w, w_{INR})$ is expressed by a product of a Gaussian and a Dirac delta distribution, which can be alternatively be seen as a single Gaussian distribution with precision $\gamma \to +\infty$ for variates corresponding to $w$ and zero covariance between $w$ and $w_{INR}$ terms by assumption (eq. 3). Concerning the weights and biases that directly parameterize the "main" network (i.e. the product $w \circ \xi$), we note that these are in general non-Gaussian, even under LA assumptions. The INR $f_{w_{INR}}(\cdot)$ transforms the (approximately) Gaussian $w_{INR}$ into a non-Gaussian density $q(\xi)$. This is multiplied by deterministic $w$ where the result follows a density that is a scaled version of $q(\xi)$. The first and second moments are equal to $W\mathcal{E}\{\xi\}$ and $W\mathcal{V}\{\xi\}W$, where $W = diag\{w\}$ and $\mathcal{E}\{\cdot\}, \mathcal{V}\{\cdot\}$ denote expectation and covariance respectively. Once we have computed a posterior over the weights, we can estimate the predictive (eq. 1) by acquiring $\xi$ samples by first sampling $w_{INR} \sim q(w_{INR})$ and evaluating $\xi = f_{w_{INR}}(\cdot)$. We finally scale them by $w$, then the product is used to compute $g(x)$ and $p(y|g(x))$ in a Monte Carlo fashion.

Alternatively, the predictive distribution (eq. 1) can be computed in closed form, as long as we impose a linearizing assumption over the network output. Specifically, this involves a first-order Taylor expansion of network output $g(\cdot)$ around $w_{INR}$. As by LA assumption, parameters $w_{INR}$ are *a posteriori* Gaussian-distributed, a linear transformation over them through linearization would result in a Gaussian predictive as well; linearization over other variables $(w, \xi)$ would not have been fruitful due to their being non-Gaussian. Hence, we only require parameters $w_{INR}$ to vary in this approximation, while we assume the rest of the parameters $w$ to be constant at their MAP solution. Formally we write:

$$g^{lin}(x) \approx g_{\bar{w}, \bar{w}_{INR}}(x) + J_{w_{INR}}(x)(w_{INR} - \bar{w}_{INR}), \tag{6}$$

where we used $J_{w_{INR}}(x) = \frac{\partial g_{\bar{w}, w_{INR}}(x)}{\partial w_{INR}}|_{\bar{w}_{INR}}$. For the predictive we then have:

$$p(y|D, x^\star) = \mathcal{N}(g_{\bar{w}, \bar{w}_{INR}}(x^\star), J^T_{w_{INR}}(x^\star)\Lambda^{-1}J_{w_{INR}}(x^\star)). \tag{7}$$

**Stochastic Weight Averaging.** An alternative over LA is to use SWAG (Maddox et al., 2019) over INR parameters. In this context, this amounts to approximating $p(w_{INR}|D)$ by a Gaussian $q(w_{INR})$ as in eq. 4, but with inverse $\Lambda$ equal to the sample covariance over the SGD trajectory:

$$\Lambda^{-1} = \frac{1}{T-1} \sum_{i=1}^{T} (w^{(i)}_{INR} - \bar{w}_{INR})(w^{(i)}_{INR} - \bar{w}_{INR})^T, \tag{8}$$

where $\{w^{(1)}_{INR}, w^{(2)}_{INR}, \ldots, w^{(T)}_{INR}\}$ are training updates of INR parameters. The predictive distribution is estimated by Bayesian model averaging through Monte Carlo sampling. Formally we have:

$$p(y|D, x^\star) \approx \frac{1}{K} \sum_{k=1}^{K} p(y|g_{\bar{w}, \xi_k}(x^\star)), \tag{9}$$

where $K$ samples $\{\xi_1, \xi_2, \ldots, \xi_K\}$ are drawn from the approximate posterior $q(\xi)$ by evaluating $w_{INR} \sim q(w_{INR})$ as described in the previous paragraph.

**Normalizing Flows.** Normalizing Flows are another poweful modeling choice for $q(w_{INR})$. In this context, $q(w_{INR})$ is freed from the Gaussian restriction and can be any parameterized flexible parametric distribution. A normalizing flow transforms an initial random variable $z$, typically sampled from a standard Normal, by applying a chain of invertible parameterized transformations.

The RealNVP model (Dinh et al., 2016) is based on a flow composed of a series of affine coupling layers defined as: $y \to m \circ z_{i-1} + (1-m) \circ (z_{i-1} \exp(s(m \circ z_{i-1})) + t(m \circ z_{i-1}))$ where $s$ and $t$ stand for scale and translation, which are typical linear mappings, while $m$ is a channel-wise masking scheme. The flow parameters can be computed by directly optimizing the variational lower bound:

$$L(w, w_{INR}) = \mathbb{E}_{q(w_{INR})} \log p(y|g_{w,w_{INR}}(x^\star)) - KL(q(w_{INR})||p(w_{INR})), \qquad (10)$$

where the carefully designed coupling layers ensure that the inverse and the Jacobian of the determinant of each transformation can be efficiently computed. The predictive distribution is estimated by Bayesian model averaging through Monte Carlo sampling similar to eq.9.

Table 1: Numerical results for classification on CIFAR10 (top) and Corrupted CIFAR10 (bottom) for different design choices. Log-Likelihood ($\uparrow$) and Expected Calibration Error ($\downarrow$) are reported.

| | | Modeling | | | Noise Structure | | | Type of INR | | Noise Type | | Activation Type | |
| --- | --- | --- | --- | --- | --- | --- | --- | --- | --- | --- | --- | --- | --- |
| | w | w$\xi$ | $\xi$ | Rank-1 | Channel | Full | Individual | Shared | Mult | Add | ReLU | Sine |
| LL | -1.29 | **-0.37** | -0.44 | -0.47 | -0.40 | **-0.37** | **-0.29** | -0.37 | **-0.37** | -5.28 | -0.289 | **-0.287** |
| ECE | **0.01** | 0.05 | 0.06 | 0.06 | **0.05** | 0.05 | 0.06 | **0.05** | **0.05** | 0.19 | **0.032** | 0.034 |
| LL | -1.80 | **-0.97** | -1.60 | -1.43 | -1.18 | **-0.97** | **-0.90** | -0.97 | **-0.97** | -6.29 | -0.50 | **-0.47** |
| ECE | 0.17 | **0.11** | 0.20 | 0.20 | 0.15 | **0.11** | **0.10** | 0.11 | **0.11** | 0.26 | **0.06** | 0.05 |

## 4 EXPERIMENTAL RESULTS

In this Section, we provide numerical results for the proposed INR-based scheme, in comparison to recent Bayesian inference methods. Namely, we compare ourselves versus the following methods: MC Dropout (Gal & Ghahramani, 2016); Bayes by Hypernet (BbH) (Pawlowski et al., 2017) Deep Ensembles (Lakshminarayanan et al., 2017) – considered among the state-of-the-art methods for uncertainty estimation in Deep Learning (Ovadia et al., 2019; Ashukha et al., 2020) – and last layer Laplace approximation (LL). We start by experimenting with different types of modeling choices and evaluate each one on a baseline classification task, in order to quantify how our method performs under different modeling scenarios. For our main numerical analysis, we deployed three different experimental setups. First, we evaluate our predictive uncertainties for our method on a $1D$ synthetic regression task. We carried out experiments to evaluate INR performance on different types of regression UCI datasets. Last, we ran image classification trials (CIFAR100, CIFAR10 and MNIST) where we compare ResNet variants for prediction and out-of-distribution robustness. We test three variants of the proposed INR-based model, namely INR-Laplace (eq. 4,5,7), INR-SWAG (eq. 4,8,9) and INR-RealNVP (eq. 10). The three variants differ w.r.t. the approximation strategy for the posterior and the predictive (cf. Section 3). For the first two cases we compute the *full Gaussian covariance for the weight posterior* (avoiding e.g. KFAC or low-rank approximations (Daxberger et al., 2021a)). Throughout our experiments, we found that the proposed model provides good predictive uncertainties on a variety of settings, highlighting the benefits of low-dimensional Bayesian inference. Concerning implementation details of the proposed and compared models and benchmark setup in general, we have moved additional information to the Appendix (App. B).

### 4.1 DESIGN CHOICES

In this Section we carry out ablation studies that justify the particular modeling and INR architecture described in subsection 3.1 and help us understand the behavior of the hypernetwork under different settings. We numerically evaluate each different potential modeling scenario by training a ResNet-20 model at CIFAR-10 according to subsection:4.4 and evaluate its MAP solution in both in and out of distribution data. Table 1 includes the main results.

Our first ablation study aims to justify the introduction and use of $\xi$ variables i.e. we investigate how the BNNs perform with only the INR for the posterior (see Table 1 under the column "Modeling"). As the $\xi$ variables only serve to induce stochasticity, removing weights $w$ result in a model which is not able to capture any information from the training data. Furthermore, augmenting $w$ with $\xi$ results in a more sophisticated model which yields better calibrated predictions. We choose the INR hypernetwork to be shared across all the layers of the main network. Sharing the INR hypernetwork, besides being efficient, can also reduce significantly the dimensionality of $w_{INR}$, as the total $d(w_{INR})$

for the individual hypernetwork will by a multiple of the number of layers of the main network. As an example, for Wide-ResNets the magnitude of this figure can be up to hundreds of variates. Despite having less parameters, the shared version of the hypernetwork is highly comparable to its more expensive counterpart as Table 1 (column labelled as "Type of INR") indicates. Also, we introduce independent nuisance factors $\xi$ for every single weight $w$. In Table 1 (column labelled as "Noise Structure") we measure the benefits of our full-rank multiplicative noise versus other low-rank modeling options used in related works (Dusenberry et al., 2020; Louizos & Welling, 2017). In the same Table (column labelled as "Noise Type"), we can see results for evaluation of two different types of noise injection in the main model, namely multiplicative noise ("Mult") and additive noise ("Add"). The additive noise hugely underperforms where multiplicative noise factors seem to provide good and calibrated solutions. Because in the multiplicative structure during training $\nabla \xi$ depends on $W$, we argue that as $W$ is responsible for fitting the data, it can pass valuable information to the hypernetwork weights in the multiplicative case leading to significant increase in overall performance. Furthermore we find that Sine/Periodic activations – the "default" choice in Sitzmann et al. (2020) – slightly outperforms a hypernet with ReLU activations as we can see in Table 1 (column labelled as "Activation Type"), even though results are still very close.

Finally, we evaluate the effects of INR network size on uncertainty estimates. We want to measure how increasing the number of paramters of the hypernetwork will affect the predictive behavior of the model. We trained 3 different INR models, with increasing numbers of trainable parameters. Following Fort et al. 2019 and Dusenberry et al. 2020, in Figure 1 we examine the normalized

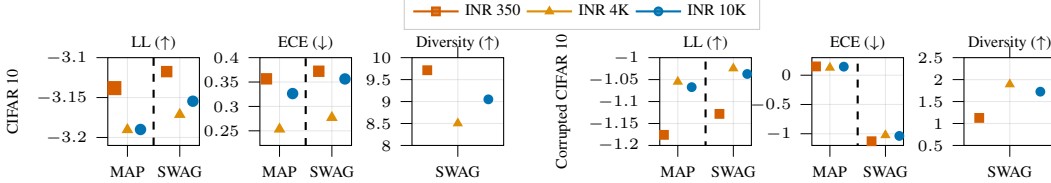

Figure 1: Comparison of Log-Likelihood ($\uparrow$), Expected Calibration Error ($\downarrow$) and Normalized Diversities between INR networks of increasing size, over CIFAR10 and corrupted CIFAR10 datasets. INR-$x$ represents an INR with $x$ parameters.

diversity of INRs of increasing size, where the posterior over $w \circ \xi$ was estimated via INR-SWAG and INR-MAP. Increasing the size of the INR hypernetwork results in more complex weight posteriors, which is depicted with better scores across all metrics in out-of-distribution data. Nevertheless, a small INR with only 350 trainable parameters is competitive in this training setup.

## 4.2 VISUALIZING UNCERTAINTY

We use a synthetic $1D$ regression task with three disjoint clusters of input data as proposed in Izmailov et al. (2020). This dataset is carefully designed to test "in-between" uncertainty, i.e. model confidence in between these disjoint clusters of data (Foong et al., 2019). Ideally, we want a model to predict high uncertainty values as test data move away from the observed data. In this test, we

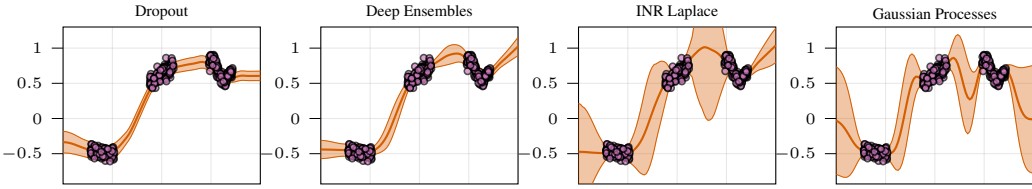

Figure 2: Visualization of the predictive distribution for the "toy" regression task. The data are denoted as purple circles, predictive mean is the solid orange line and the shaded region is $\pm$ 1 std.

use a fully-connected architecture with hidden layers that have $[200, 50, 50, 50]$ neurons respectively. Following Izmailov et al. (2020), the network takes two inputs $\hat{x} = (x, x^2)$ and outputs a single

real value $y = f(\hat{x})$. The INR network has 3 layers consisting of $[2, 10, 4]$ neurons respectively, resulting totally in 160 training parameters (equal to only $1\%$ of the number of the $\xi$ parameters, cf. Section 3). Results are shown in Figure 2. We also include a Gaussian Process (GP) with a Radial Basis Function (RBF) kernel as the state of the art for this problem. Our INR-Laplace preserves more of the uncertainty regarding both "out" and "in-between" of the observed data. Other methods, like Deep Ensembles and MC Dropout infer a desirable uncertainty structure but still remain quite overconfident. Furthermore, the proposed INR model is able to maintain the appealing characteristics of the approximate inference methods applied, specifically the stationary structure (or in-between-uncertainty) benefits of the Linearized Laplace approximation as shown in multiple works (Kristiadi et al., 2020; Daxberger et al., 2021b).

## 4.3 UCI REGRESSION

We next test our method on the UCI regression tasks (Asuncion & Newman, 2007). We experiment with 8 UCI regression datasets using standard training-evaluation-test splits from Hernández-Lobato & Adams (2015) and their GAP-variants (Foong et al., 2019). To measure performance we deployed Gaussian test log-likelihood (LL). Our training strategy follows the work of Daxberger et al. (2021b). The INR network has 4 layers consisting of $[5, 5, 5, 1]$ neurons respectively, resulting totally in 70 training parameters (equal to only $2\%$ of the number of the $\xi$ parameters, cf. Section 3)

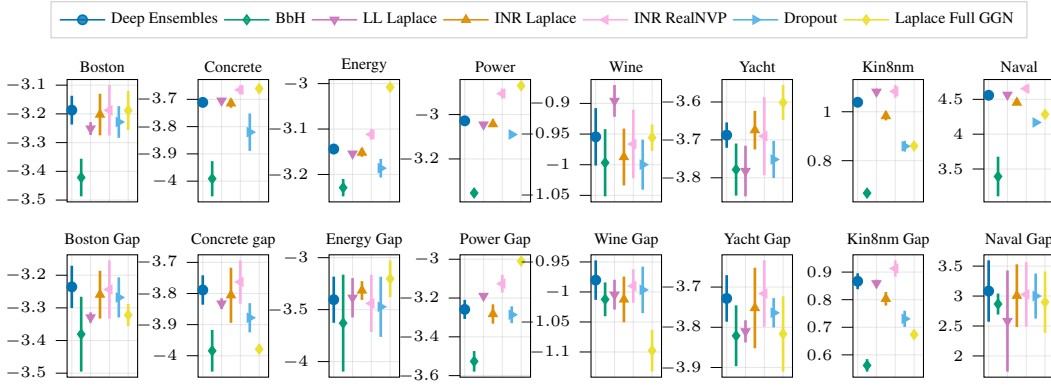

Figure 3: Numerical results for regression trials on UCI datasets (Asuncion & Newman, 2007). Mean values of test Log-Likelihood ($\uparrow$) are shown with $\pm$ 1 standard deviation error bars, obtained over standard (Hernández-Lobato & Adams, 2015) and GAP (Foong et al., 2019) splits.

The main results are depicted in Figure 3. The small MLP network enabled us to compute the full GGN matrix in the Laplace approximation of the main network and add it as baseline. As we can see, INR combined with RealNVP or LA achieves better test log-likelihood – a metric which considers both uncertainty and accuracy – compared to BbH and LL Laplace approximation, while followed closely by MC Dropout. Furthermore, the proposed INR remain competitive with Deep Ensembles networks, even surpassing them in five out of eight datasets while overall being close enough, in both standard and gap splits, as standard deviation bars indicate.

## 4.4 IMAGE CLASSIFICATION UNDER DISTRIBUTION SHIFT

We evaluate our method on standard image classification tasks over the CIFAR10, CIFAR100 (Krizhevsky et al., 2009) datasets. We use ResNet-50 (He et al., 2016) in order to test the ability of the proposed INR-based method to scale into larger models. A capable Bayesian inference technique is critical when applied in deep models, as they tend to exhibit less accurate calibration in this context (Guo et al., 2017). We provide experiments in a context of high degree of distribution shift, as under these conditions the evaluation of predictive uncertainty is the most useful in practice (Ovadia et al., 2019). Our INR hypernetwork (Sitzmann et al., 2020), has 4 layers with $[10, 10, 10, 1]$ neurons each, resulting in 260 training parameters (only $0.001\%$ of the parameters $\xi$). Following Ovadia et al. (2019); Antorán et al. (2020), we train ResNet50 on CIFAR10/CIFAR100 and evaluate on data subject to 16 different types of corruption with 5 levels of noise intensity each (Hendrycks &

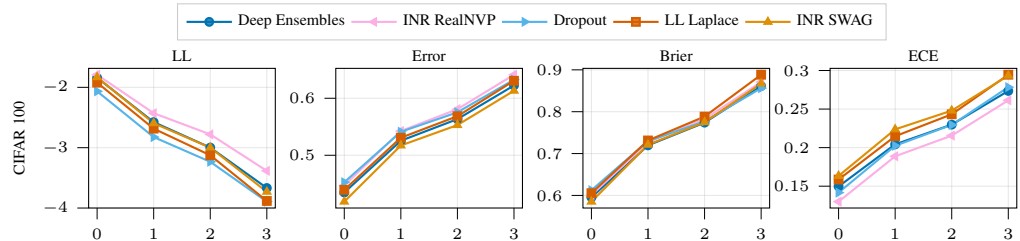

Figure 4: Numerical results for classification trials on Corrupted CIFAR100 dataset. The $x$-axis of each plot corresponds to increasingly corruption levels.

Dietterich, 2018). As Fig. 4 indicated, one of the proposed variants, INR-RealNVP, outperforms non-INR methods in terms, log-likelihood and expected calibration error. Both INR-based methods outperform LL Laplace and MC Dropout which are overconfident in their predictions and more often erroneous while still being competitive w.r.t Deep Ensembles. Overall, these results suggest that the proposed approach produces more calibrated and accurate models than other popular uncertainty quantification approaches.

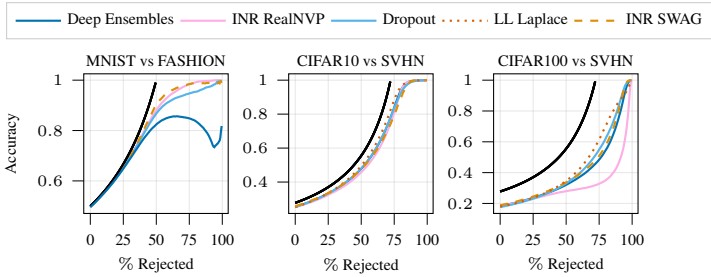

Figure 5: Rejection-Classification plots.

We quantify the quality of uncertainty estimates by jointly evaluating the predictive entropy of our model on an in-distribution and an OOD test set. Ideally, we want predictive entropy to be high on OOD data as predictions should be more uncertain, and vice versa. Following Antorán et al. (2020); Nadeem et al. (2009), we deployed the OOD rejection scenario by jointly evaluating the entropy of our model on an in-distribution and OOD test set, where we allow the models to reject an increasing proportion of the data based solely on their entropy values. Ideally, we want highly calibrated and robust models to reject all the OOD examples, as well as the in-distributional examples when the corresponding predictions are inaccurate. Figure 5 illustrates on what percentage of the remaining non-rejected examples the predictions are accurate. On CIFAR10-SVHN all methods have the same performance, while on CIFAR100 the INR-RealNVP model fails to distinguish very uncertain in-distribution data from low uncertainty OOD ones. On MNIST-Fashion, the proposed methods INR-SWAG and INR-RealNVP perform best, followed by LL Laplace and Dropout.

Finally, we tried to measure the quality of proposed low-dimensional spaces in terms of predictive uncertainty. Specifically, we compare our INR low dimensional space with: rank-1 (Dusenberry et al., 2020) Wasserstein subnetwork (Daxberger et al., 2021b) and partially stochastic Resnets from Sharma et al. (2023). We trained (each method) combined with a Resnet18 for 100 epochs in CIFAR100 while keeping the approximate inference method the same across all low-dimensional spaces. Results in Table 2 show a trend in favor of both proposed INR-$x$ methods and validate to a considerable degree the premise of our method: instead of choosing a subset or subnet following the rationale of the corresponding methods, the INR produces $\xi$ outputs that endow the full network with the desirable stochasticity, while keeping the dimensionality of the random process that we want to do inference upon at a low level.

## 5 RELATED WORK

**Hypernetworks.** Hypernetworks are NNs that are used to predict deterministically the parameters of another, typically larger network, termed the "primary" network. The terminology is due to Ha et al.

Table 2: Numerical results for classification trials on CIFAR100 for different proposed low-dimensional spaces alongside their inference time.

| Subspace | Inference | Standard | | | | Corrupted | | | | Time ↓ |
|---|---|---|---|---|---|---|---|---|---|---|
| | | LL ↑ | Error ↓ | Brier ↓ | ECE ↓ | LL ↑ | Error ↓ | Brier ↓ | ECE ↓ | |
| Rank1 | SWAG | −2.29 | 0.34 | 0.55 | 0.22 | −4.77 | 0.57 | 0.92 | 0.39 | 0.28 |
| | Laplace | −4.01 | 0.31 | 0.97 | 0.66 | −4.25 | 0.58 | **0.97** | 0.40 | 0.55 |
| INR | SWAG | **−2.09** | **0.30** | 0.50 | 0.22 | −4.18 | 0.53 | 0.84 | 0.36 | **0.11** |
| | Laplace | **−3.91** | **0.30** | **0.96** | 0.67 | −4.19 | 0.58 | **0.97** | **0.39** | 0.51 |
| Subnetwork | SWAG | −2.14 | **0.30** | **0.49** | **0.20** | **−3.97** | **0.51** | **0.82** | **0.34** | 0.29 |
| | Laplace | −3.95 | 0.32 | **0.96** | 0.65 | −4.13 | **0.51** | **0.97** | 0.46 | **0.42** |
| Partially Stochastic | SWAG | −2.14 | **0.30** | **0.49** | **0.20** | **−3.97** | **0.51** | **0.82** | **0.34** | 0.28 |
| | Laplace | −3.99 | 0.34 | 0.97 | **0.63** | −4.18 | **0.51** | 0.98 | 0.47 | 0.49 |

(2016), however the main idea can be traced back to earlier works (see discussion in e.g. Krueger et al. 2017; Karaletsos et al. 2018). Krueger et al. (2017) have been among the first to extend hypernetworks to a Bayesian setting. Their Bayesian hypernetwork, modelled as a normalizing flow, learns to predict distributions of weights for the primary network. The flow predicts scaling per-neuron factors for the primary network weights. This is similar to the closely related (Louizos & Welling, 2017), which however require an extra inference network to estimate the entropy term of the VLB. Almost concurrently, Pawlowski et al. (2017) proposed BbH for VI with implicit distributions. They use a discriminator network for density ratio estimation (DRE) in the context of prior-constrastive VI (Huszár, 2017), and a generator to model the variational distribution. Shi et al. (2017) use a kernel method for DRE instead of a discriminator. Karaletsos et al. (2018) and Karaletsos & Bui (2020) explore hierarchical prior modeling using NN-based implicit distributions and Gaussian processes. INRs have also been used for approximating model parameters of deep NNs (Romero et al., 2021a;b).

**Low-Dimensional Inference**. Bayesian inference in a low-dimensional space is another important related concept, with often considerable overlap to works that can be understood as forms of hypernetworks. Dusenberry et al. 2020, in the spirit of Wen et al. 2020, employ rank-1 multiplicative noise components, before attempting to estimate an approximate posterior over the weights. Izmailov et al. 2020 adopt *post-hoc* Bayesian inference by constructing a subspace of the BNN weights. They apply high fidelity inference on these small subspaces, and were able to produce state-of-the-art results at a moderately low computational cost. Pradier et al. 2018 learn a non-linear latent representation of network weights. Another subgroup of related work can be described as selecting a portion of the BNN parameters to be treated as random variables, and leaving the rest of the model to work deterministically. One of the most popular and straightforward approaches are last-layer BNNs. By selecting *a priori* only the last layer to have a probabilistic treatment, they resort to a linear model which ensures analytical tractability of both inference and predictive distribution in the spirit of Gaussian processes, while the remaining NN structure acts as a feature extractor (Watson et al., 2021; Snoek et al., 2015; Lázaro-Gredilla & Figueiras-Vidal, 2010; Weber et al., 2018). Finally, Daxberger et al. 2021b first obtain a MAP estimate of all weights, then define a subnetwork selected in a way that aims to maximally preserve predictive uncertainty. The small size of the subnetwork allows for the use of a full-covariance Gaussian posterior in tandem with linearized LA (MacKay, 1992).

**Stochastic INRs.** INRs have been used as models for signal compression (Dupont et al., 2021a), and more recently they have been extended to the Variational Bayesian setting (Guo et al., 2023). Shen et al. (2021) extend NeRFs to learning distributions of all possible radiance fields. A simple variational posterior is assumed, and the base model is extended to learn uncertainty estimates over scene parameters. Vasconcelos et al. (2022) use a BNN as an INR of computerized tomography.

## 6 CONCLUSION AND FUTURE WORK

We have presented an approach for scalable and efficient Bayesian Deep Learning, that leverages on the small size and representational strength of INRs. Our claims are corroborated by the reported experimental results, which show that the integration of the proposed method results in improving considerably overall uncertainty estimates. For future work, we aim at exploring other ways to integrate INRs (e.g. multiplicative filter networks (Fathony et al., 2020)) as well as integrating with different types of approximations, such as Hamiltonian Monte Carlo (Neal et al., 2011).

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

APPENDIX

## A INR HYPERNETWORK DETAILS

This section delves deeper into the INR hypernetwork. We analyze its functionality and provide a graphical illustration of the process in Fig. 7. Additionally, we present the training and inference procedures of our proposed low-dimensional inference scheme in two separate algorithms outlined in Figure 6. .

---

**Algorithm 1** Training procedure

**Require:** $I$ (Indices of main network weights), *Net* (Main network), *INR* (INR hypernetwork), *Dataset*.
  **for** each Epoch **do**
    **for** $(x, y)$ in *Dataset* **do**
      $y^{\star} = Net(x, \xi)$
      loss $= (y, y^{\star})$
      update *INR* w.r.t loss
      update *Net* w.r.t loss
    **end for**
  **end for**

---

**Algorithm 2** Inference procedure

**Require:** $I$ (Indices), *Net* (Main network), *INR* (INR hypernetwork), *Testset Approximate Inference* (Approximate inference method) *MC Samples* (Number of Monte Carlo samples).
  **for** $x$ in *Testset* **do**
    **for** $j$ in range *MC Samples* **do**
      $\xi_j = Approximate\ Inference(INR, I)$
      $y^{\star} = Net(x, \xi_j)$
    **end for**
  **end for**
  Calculate $y^{\star}$ statistics

---

Figure 6: High level pseudo-code to introduce our method's behavior in training and inference settings (in this setting, a post-training Monte Carlo-based approximate inference method is implied).

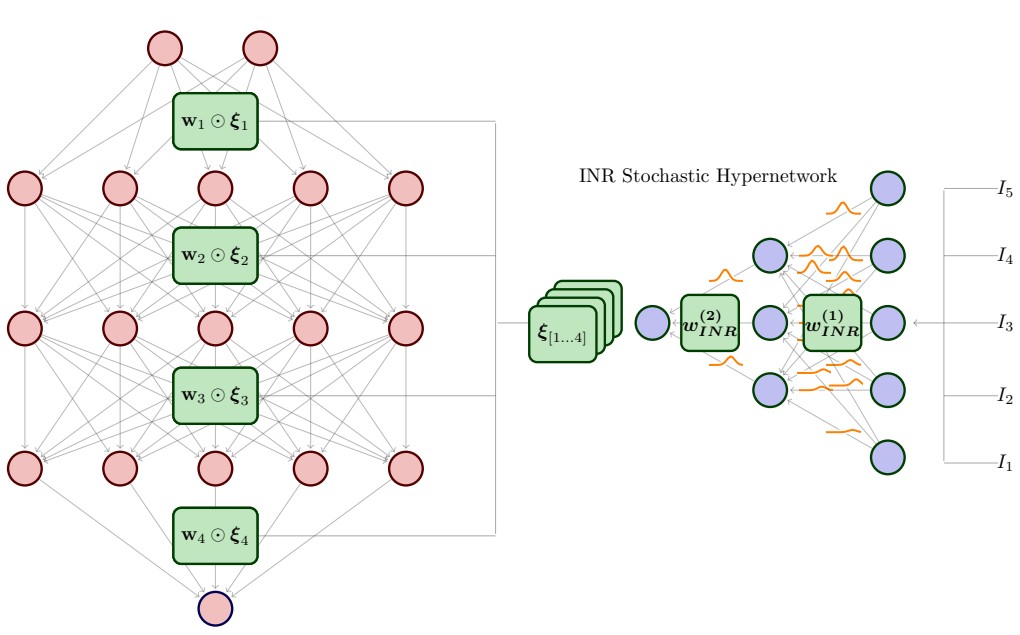

Figure 7: Illustration of the proposed INR model.

As for the weight coordinates $I$, in practice these values are batched and computed separately for each layer. For the $i-th$ layer indices/input-coordinates positions have the shape $[n, I_{dims}]$ where $n$ is the number of the total main network parameters of the $i-th$ layer and $I_{dims}$ is the dimensionality of the indices. For example, for a convolutional main layer $I_{dims} = 5$, the first 4 positions correspond to the kernel weights plus 1 dimension to act as the layer position (conditional position for each layer).

## B EXPERIMENTAL SETUP

In this Section, we provide all the experimental details that were used in order to produce the results of the main paper. Our experimental setups and procedures are heavily influenced by current practices in the literature. In all experiments we chose the SIREN (Sitzmann et al., 2020) network to serve as the INR hypernetwork, due its popularity and its ability to model highly complex signals without any use of positional encoding layers (for the coordinate tensors $I$). Furthermore, Sitzmann et al. (2020) described a hypernetwork initialization scheme for the $w_{INR}$ parameters, that results in values $\xi$ that are initially Normally distributed. This is in general beneficial for the training procedure and it is also common practice for initializing multiplicative noise (Kingma et al., 2015; Gal & Ghahramani, 2016).

### B.1 DESIGN CHOICES.

For each different modeling scenario we trained a ResNet-20 (He et al., 2016) on the CIFAR10 dataset. All models are trained using the Adam optimizer with learning rate equal to $10^{-3}$, weight decay equal to $10^{-6}$ and a batch size equal to 256 running for 100 epochs. We evaluated the MAP solution for each model in clear and corrupted test data. As for evaluating the size of the INR hypernetwork we also deployed SWAG. We used a full Gaussian covariance to approximate the distribution of $w_{INR}$, and used 10 epochs of average with a learning step of 0.01. We evaluated the effect of the increasing size of hypernetworks by using the Log-likelihood, Expected Calibration Error and the Normalized Diversity. Concerning the latter, a typical way to quantify diversity is to compute the fraction of points where discrete predictions differ between two members, averaged over all possible pairs. This disagreement measure is normalized by $(1 - accuracy)$ of each prediction to take into account its sample predictive accuracy. Recent works (Pang et al., 2019; Fort et al., 2019) point out that measuring the diversity of individual predictions obtained from each sampled network can highlight the quality of uncertainty.

### B.2 VISUALIZING UNCERTAINTY.

This visualization task is highly suited to quantify "in-between" uncertainty of a model, as recent works found that standard numerical evaluation metrics such as log-likelihood struggle to fully capture this behavior, while at times overconfident methods may obtain better scores (Yao et al., 2019; Ashukha et al., 2020). We train a single, 2 hidden layer network, with 50 hidden ReLU units per layer using MAP inference until convergence. For the INR network $f_{w_{INR}}(\cdot)$ we used a SIREN Sitzmann et al. (2020) The INR network has 3 layers consisting of $[2, 10, 4]$ neurons respectively, resulting totally in 160 training parameters. Concerning the hyperparameters we used $\Omega_1 = 30$ for the first INR layer and $\Omega_l = 1$ for the rest while keeping the the parameter $c = 1$ fixed for all layers. The INR weights $w_{INR}$ are initialized uniformly as $\sim \mathcal{U}(-\sqrt{c/n}/\Omega, \sqrt{c/n}/\Omega)$. The input coordinates $I \in \mathbb{R}^2$ are normalized to be in the range $[-1, 1]^2$. We optimize the Gaussian log-likelihood of our data, where the mean is produced by the network and the variance is a hyperparameter learnt jointly with NN parameters. We used a full batch Adam optimizer with a learning rate of $10^{-3}$, $\alpha = 0.9$, $\beta = 0.999$ and weight decay $= 10^{-4}$. We trained all models for 600 epochs (since the amount of training samples are less than the actual training parameters). We used the same strategy for all of the baselines. We deployed deep ensembles with an ensemble of 5 networks, as suggested by Ovadia et al. (2019). Dropout was set with dropping probability of 0.1. For the INR-RealNVP, following the literature, we tempered the posterior by applying a weight on the Kullback-Leibler term of the ELBO, equal to 0.1. For the INR equipped with the linearized Laplace we set the prior precision of $\lambda = 0.001$, where $C = \lambda^{-2} I_{d_\xi}$. Methods that required Monte Carlo sampling for estimating the predictive distribution use 30 MC samples during testing and 1 sample during training.

### B.3 UCI.

We experiment with 8 UCI regression datasets using standard training-evaluation-test splits from Hernández-Lobato & Adams (2015) and their gap versions (Foong et al., 2019). In this test, we use a fully-connected architecture with hidden layers that have $[50, 50, 20]$ neurons respectively followed by ReLU activation. All the training details are applied to all the regression datasets regardless of their individual characteristics such as size, input dimensions, etc. We used homoscedastic regression

methods with a global variance parameter, $\mathcal{N}(y_i, g_w(x_i), \hat{\sigma}^2 I)$, where the logarithm of global log-variance $\log\hat{\sigma}^2$ (in order to ensure positivity) is jointly optimized with the model parameters. Our training strategy follows Daxberger et al. (2021b). We trained all methods for 50 epochs, except INR-RealNVP, which needed approximately 5 additional epochs to adapt. We employed early stopping if validation performance does not increase for 10 consecutive epochs. The weight settings which provide best validation performance in terms of log-likelihood are kept for testing. Again we used the Adam optimizer with a learning rate equal to $10^{-3}$, weight decay equal to $10^{-6}$, and a batch size equal to 512 samples. For the INR equipped with SWAG, we used full Gaussian covariance to approximate the distribution of $w_{INR}$, and used 25 epochs of average with a learning step of 0.01. For INR-Laplace, after trying several precision values we use a prior with a precision value $\lambda = 0.005$, as it yielded better validation results across all datasets.

### B.4 IMAGE EXPERIMENTS.

Through our image experiments we deployed the ResNet50 architecture (He et al., 2016). As it is common practice, we applied several modifications to the original architecture such as replacing the kernel size of the first strided convolutional layer ($7 \times 7$) to size $3 \times 3$. Additionally, we remove the first max-pooling layer. The rest of the ResNet details were set according to Goyal et al. (2017). For the INR network $f_{w_{INR}}(\cdot)$ we used a SIREN (Sitzmann et al., 2020) shared across each layer of the main network. We used a variety of metrics, these include: test log-likelihood (LL); Brier score (Brier et al., 1950), which is a metric that measures accuracy of predictive probabilities by computing their mean squared distance from the one-hot class labels; the Expected Calibration Error (ECE, Naeini et al. 2015), a metric which measures the difference between predictive confidence and empirical accuracy in classification. A detailed explanation of uncertainty evaluation metrics can be found in Antorán et al. (2020); Ashukha et al. (2020); Ovadia et al. (2019). In our experiments we emphasized on out-of-distribution performance, as model that was well-calibrated on the training and validation distributions must ideally remain so on shifted data. Regarding the completely "out-of-distribution" (OOD) data, we expect the empirical entropy of the predicted distribution to be quite high. Essentially, a good model must be uncertain according to the degree that test inputs are far from the training distribution. For Dropout experiments, we add Dropout to the standard ResNet model in between the $2^{nd}$ and $3^{rd}$ convolutions in each ResNet block (Ashukha et al., 2020). We used an ensemble of 5 elements for prediction. Ensemble elements differ from each other in their initialization, which is sampled from the He initialization distribution (He et al., 2015). All models are trained using the Adam optimizer with learning rate equal to $10^{-3}$, weight decay equal to $10^{-6}$, with a batch size equal to 256 running for 50 epochs for MNIST and 150 epochs for the CIFAR10/CIFAR100 experiments respectively. The weight settings which provide best validation performance in terms of log-likelihood are kept for testing. During training, we also used plain data augmentation strategies including random image cropping and random horizontal flips. We used INR-SWAG for 10 epochs with learning rate equal to $10^{-4}$. For INR-RealNVP, the base Gaussian distribution is set to $\mathcal{N}(0, 0.1I)$, transformed with a cascade of 4 coupling layers. Finally as for the experiments validating the uncertainty quality per low dimensional space we trained (each method) combined with a Resnet18 for 100 epochs in both Cifar10 and Cifar100 datasets while keeping the approximate inference method fixed same across all low dimensional spaces. While for each subspace method we followed the hyperparameters proposed in the original papers, for SWAG and Linearized Laplace with GGN, in order to be able to run across low dimensional spaces we choose the covariance to have Diagonal structure. We used SWAG for 10 epochs with learning rate equal to $10^{-3}$. For the Laplace, we use a prior with a precision value $\lambda = 1.0$. All hyperparameters stayed the same across each method for comparison. Inference time (Table 2) for Resnet18 combined with different stochastic subspaces and different approximate inference methods was measured in seconds and for a batch of 10 CIFAR images.

## C ReLU AND SINUSOIDAL HYPERNETWORKS

This section delves deeper into the activation function used in the hypernetworks. Our ablation study, focusing on SIREN activation, suggests that the hypernetworks need to model high-frequency representations of the weight perturbations. We begin by empirically quantifying the benefits of each activation type by evaluating the performance of the Maximum A Posteriori (MAP) estimate.

We trained Resnet18 in both CIFAR10 and CIFAR100 for 100 epochs to evaluate the predictive capabilities of the Sinusoidal hypernetwork versus each ReLU counterpart.

Table 3: Numerical results for classification trials with different hypernetwork activations.

| Dataset | Hypernet Activation | Accuracy ↑ | LL ↑ | Error ↓ | Brier ↓ | ECE ↓ |
|---------|---------------------|------------|------|---------|---------|-------|
| CIFAR10 | ReLU | 91.11 | −0.48 | 0.08 | 0.14 | 0.05 |
| | Sine | 91.70 | −0.44 | 0.08 | 0.13 | 0.05 |
| CIFAR100 | ReLU | 67.79 | −2.54 | 0.32 | 0.53 | 0.23 |
| | Sine | 68.49 | −2.39 | 0.31 | 0.52 | 0.22 |

In Table 3 we find that Sine/Periodic activations (the "default" choice in SIREN) slightly outperforms a hypernet with ReLU activations. Still, results are very close, though there is a trend in favor of sine in both benchmarks. The original motivation behind using the sine activation is related to modeling high-frequency content, which translates as details in structured signals such as images or video Sitzmann et al. (2020). We can however see this "in the top of its head", so to speak: in structured signals we care more for low-frequency content, and high-frequency is a "good-to-have" content. We can interpret an input semantically if we see its low frequencies, but not necessarily vice versa. For example, image compression will invariably throw away high frequencies first, and the last frequencies to lose will be the lower ones. Our conjecture is as follows: When using an INR to model perturbations, we are faced with a different situation, that corresponds to a different "frequency landscape" (perhaps even different than the one of model weights). In particular, we do not have any reason to differentiate lower or higher frequency content in any respect. We "care" for all frequencies, so we need to have a good way to model high frequencies as well. Perhaps this is the reason the sine activation gives a small edge over ReLU.

To elaborate further on this argument, we constructed a setting where we can visualize the $\xi$ parameters and see if we can observe any meaningful connection between hypernetwork activation and frequencies modelled by the hypernetwork.

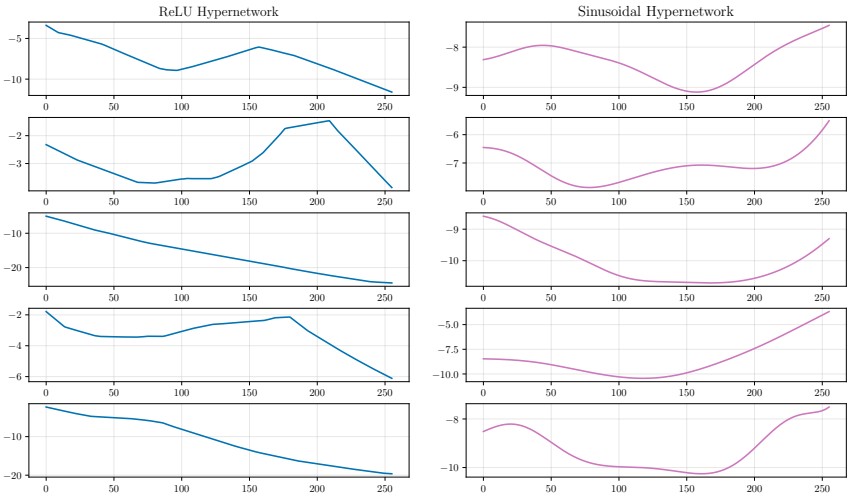

Figure 8: Values of $\xi$ as a function of input weight coordinates (channel-wise).

In Figure 8 we plotted the the values of $\xi$ as a function of input weight coordinates. Specifically for Resnet18 trained on CIFAR we plotted the flattened values for each specific kernel position across channels (channel slice) for 2 different convolutional layers. Both types of hypernetworks produce well structured perturbation functions. The $\xi$ values produced from the sinusoidal hypernetwork are expressed as a somewhat oscillatory behavior w.r.t. channel position, which translates as higher frequency content. As for the ReLU perturbations, while having some high frequencies due to the discontinuity of the ReLU activation, the overall signal has a smooth structure less complicated that

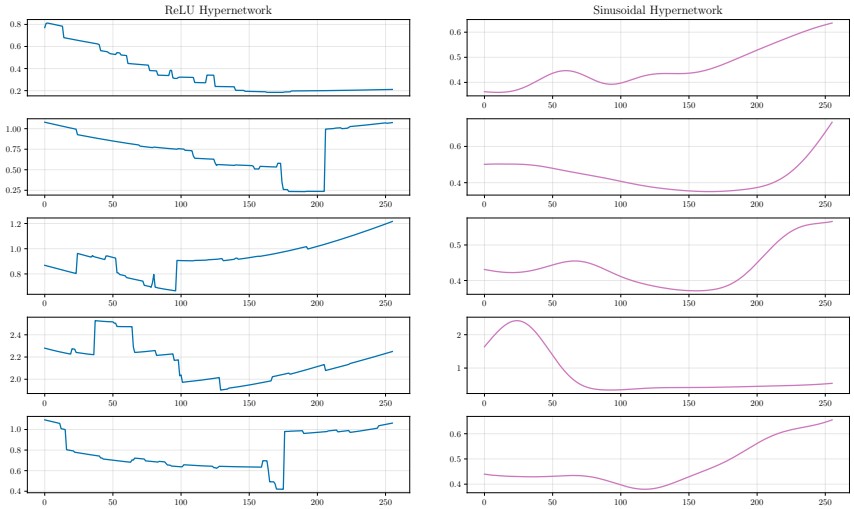

Figure 9: Empirical variance of $\xi$ as a function of input weight coordinates (channel-wise).

the sinusoidal ones in some cases. Unsurprisingly, the ReLU result consists of practically piecewise linear components. This is what we believe that highlights the marginally better performance of SIREN hypernetworks. Furthermore, following the same experimental procedure we plotted alongside the mean values of $\xi$ also their variance (Fig. 9), as this was computed from the SWAG-diagonal approximate inference method, again as a function of channel coordinates. We can observe that the variance has the same structural properties as the mean values of $\xi$. Thus, we believe that it makes sense for the main network convolutional kernel to take advantage of its structure.

## D    EVALUATING INR HYPERNETWORK SIZE

We added an ablation w.r.t. INR size following the UCI regression setting in our method 4. We compare four different versions of INR hypernetworks with an increasing number of parameters each, namely (BIG=2500 MED=625, SMALL=75,XSMALL=10), all combined with a Full GGN Laplace approximate posterior. From the experiments (Fig. 10) we can observe that there is a

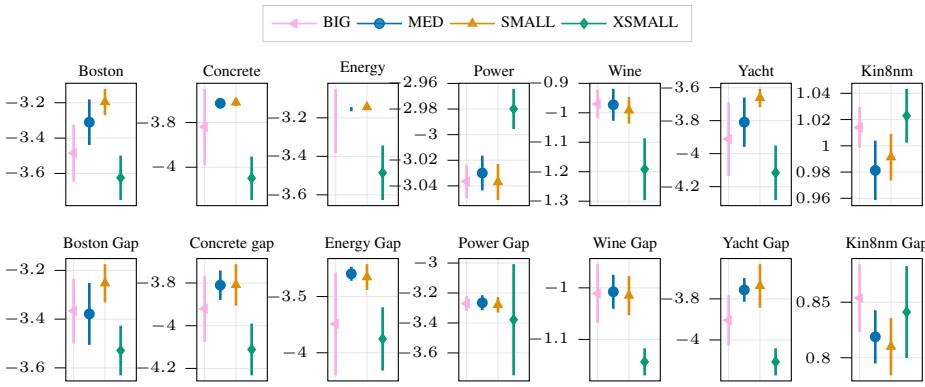

Figure 10: Numerical results for regression trials on UCI standard (Hernández-Lobato & Adams, 2015) and GAP (Foong et al., 2019) splits for different hypernetwork sizes.

limit to where one can easily scale the INR hypernetwork and simultaneously gain performance. Individual characteristics play significant role to the INR size (main network size, dataset size, dataset dimension).

# E COMPUTATIONAL TIME

Regarding the computational time requirement of our method, it can be decomposed as follows:

$$t_{\text{Total}} = t_{\text{Hypernet evaluation}} + t_{\text{Approximate Inference}} \qquad (11)$$

Where hypernetwork evaluation time according to Table 4 makes the overall network in practice $\approx 1.2$ slower than the vanilla network training. As for the approximate inference time although these methods we are using in our experiments are expensive, because in our method they are applied in the small dimensional INR space in general it takes less time to evaluate. In Table 5 we are considering the computational time of inference of our method versus standard inference popular ones.

Table 4: Indicative time requirements for INR-based hypernetwork model.

| Our method | | Vanilla Network | | Our method (fixed $\xi$ perturbations) | |
|---|---|---|---|---|---|
| Forward | Backward | Forward | Backward | Forward | Backward |
| $0.0069 \pm 0.0001$ | $0.014 \pm 0.008$ | $0.0046 \pm 0.0001$ | $0.011 \pm 0.000$ | $0.0045 \pm 0.0002$ | $0.009 \pm 0.001$ |

Table 5: Computational time of INR low dimensional inference versus other approximate inference methods.

| Method | Deep Ensembles | Dropout | LL Laplace | INR SWAG | INR RealNVP |
|---|---|---|---|---|---|
| Inference Time | $0.9014 \pm 0.0273$ | $0.0372 \pm 0.0066$ | $2.0030 \pm 0.0073$ | $0.6393 \pm 0.0184$ | $0.2045 \pm 0.0043$ |

For the Deep Ensembles method the obtained values include additional overhead such as ensemble element loading etc. as it is common practice. Furthermore, the Linearized LL Laplace is much slower than the other methods as computing the Jacobian for the ResNet50 reaches the limits of our computational budget at this time.

As for the overhead in terms of learnable parameters, we have: $W_{inr}$ (total number of the hypernetwork parameters), and $q_{inr}$ (number of approximate inference parameters applied on the INR space), which as we mention in the main paper is in fact much less than $q_W$ (number of approximate inference parameters applied on the full set of main network weights). Performance-wise our method is still being competitive w.r.t. methods like ensembles of $D$ networks which at best is $D$ times slower than the vanilla network.

Furthermore, because the main overhead of our method is the hypernetwork evaluation we investigated the following alternative training scheme, to further improve our method in terms of time. Instead of training the main network weights $W$ and $W_{INR}$ together we update the $W_{INR}$ parameters every 10 epochs of the main network training, this significantly reduces the computational overhead of our method and we hypothesize it can scale to ImageNet models and datasets. Inference time for Resnet18 combined with different stochastic subspaces and different approximate inference methods (time is measured in seconds and for a batch of 10 CIFAR images).

Table 6: Numerical results for classification trials of ResNet18 in CIFAR100.

| Training Scheme | Accuracy ↑ | LL ↑ | Error ↓ | Brier ↓ | ECE ↓ |
|---|---|---|---|---|---|
| Full Training | 69.01 | $-2.32$ | 0.30 | 0.51 | 0.22 |
| Alternative Training | 68.59 | $-2.38$ | 0.31 | 0.52 | 0.22 |

# F ADDITIONAL EXPERIMENTS

**Further Image Experiments.** Following Antorán et al. (2020); Daxberger et al. (2021b); Ovadia et al. (2019), we train all methods on MNIST and evaluate their predictive distributions on increasingly rotated digits. We trained the models for 50 epochs using the Adam optimizer. The results are depicted in Figure 11. The importance of distributional shift expressed in this experiment via rotation of the

original test set, which is highly informative as all methods perform more or less the same until the degradation shift reaches high intensity, where at this point methods begin to differentiate from one another. While the error of the prediction remains the same, metrics such as ECE and LL favor INR inference and Dropout which surpass the Deep Ensembles and LL Laplace as degradation increases significantly.

Table 7: Numerical results for classification trials on different proposed low-dimensional spaces (CIFAR10).

| Subspace | Inference | Standard | | | | Corrupted | | | |
|---|---|---|---|---|---|---|---|---|---|
| | | LL ↑ | Error ↓ | Brier ↓ | ECE ↓ | LL ↑ | Error ↓ | Brier ↓ | ECE ↓ |
| Rank1 | SWAG | −0.41 | 0.08 | 0.13 | 0.05 | −1.25 | 0.22 | 0.35 | 0.14 |
| | Laplace | −1.56 | 0.09 | 0.68 | 0.68 | −1.70 | 0.22 | 0.73 | 0.57 |
| INR | SWAG | −0.32 | 0.07 | 0.12 | 0.04 | −1.16 | 0.21 | 0.35 | 0.14 |
| | Laplace | −1.56 | 0.11 | 0.68 | 0.66 | −1.66 | 0.19 | 0.32 | 0.58 |
| Subnetwork | SWAG | −0.42 | 0.07 | 0.12 | 0.04 | −1.45 | 0.23 | 0.38 | 0.17 |
| | Laplace | −1.55 | 0.09 | 0.68 | 0.68 | −1.65 | 0.19 | 0.71 | 0.58 |
| Partially stochastic | SWAG | −0.42 | 0.07 | 0.12 | 0.04 | −1.44 | 0.20 | 0.38 | 0.17 |
| | Laplace | −1.56 | 0.09 | 0.68 | 0.70 | −1.67 | 0.21 | 0.72 | 0.59 |

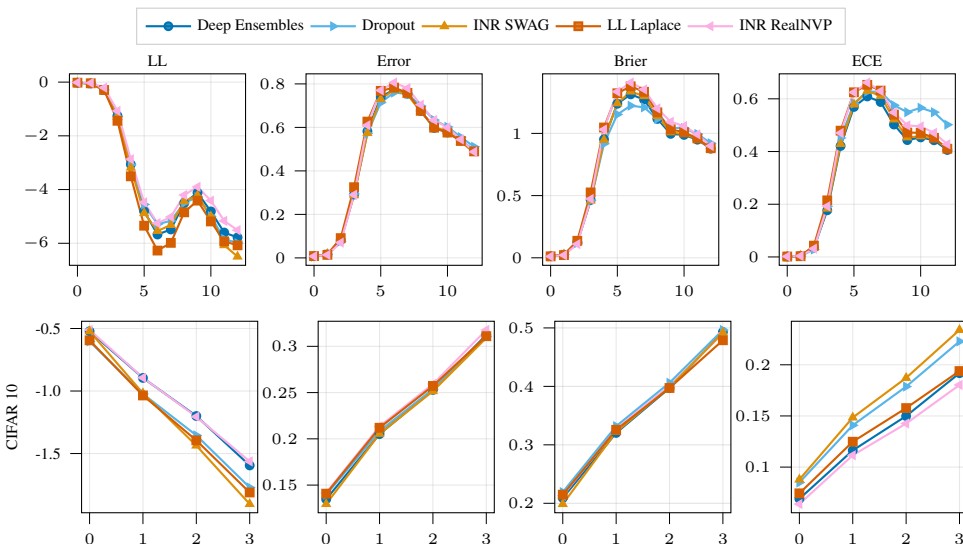

Figure 11: Numerical results for classification trials on Corrupted MNIST dataset. Log-Likelihood (↑), Expected Calibration Error (↓), Brier Score (↓), Error (↓) and Accuracy (↑) are used for comparison. The $x$-axis of each plot corresponds to increasingly levels of corruption intensity.

# G  QUALITATIVE EVALUATION OF EMPIRICAL DENSITIES

In this Section, we qualitatively inspect the approximate posterior distributions produced by INR variants in regression and classification settings. First, in Figure 12 we plot the empirical density of $w \circ \xi$ for the network trained on the toy regression task. The variables are acquired by evaluating first eq:4 with 400 samples. Then we transform each sample according to $\xi = f_{INR}(\cdot)$ and finally scale the resulted values by $w$.

As we can see, INR-based models produce non-Gaussian approximate posterior distributions. Our results are in line with works like Fortuin (2022), which analyzed the empirical weight distributions of SGD-trained networks with different architectures, suggesting that fully connected neural networks learn heavy-tailed weight distributions.

We plotted empirical covariance matrices (see Figure 15) that correspond to part of the $w \circ \xi$ parameters (specifically, the parameters that are "connected" to the first output neuron of the first layer of the main network). We can see that even the INR-based models are able to produce covariance matrices with high-magnitude off-diagonal elements. This result validates the use of more expressive posterior distributions and highlights the performance of our hypernetwork method in the previous tasks.

We evaluated the empirical densities of convolutional layers following the classification setting of subsection 4.4. More specifically, we trained a ResNet-50 using the INR-RealNVP method on CIFAR10 dataset and evaluated the approximate distribution of $w \circ \xi$ for the first convolutional layer of the network, following the same sampling procedure as before. Results are depicted in Figure 14, where the density histograms of the kernel values are Gaussian-like but still placing a lot of probability mass towards the tails.

We plotted the empirical covariance (Figure 14 left) of values belonging to the same $3 \times 3$ kernel for nine different kernels. The covariance matrices indicated high spatial correlations of kernel values as was expected (Fortuin, 2022).

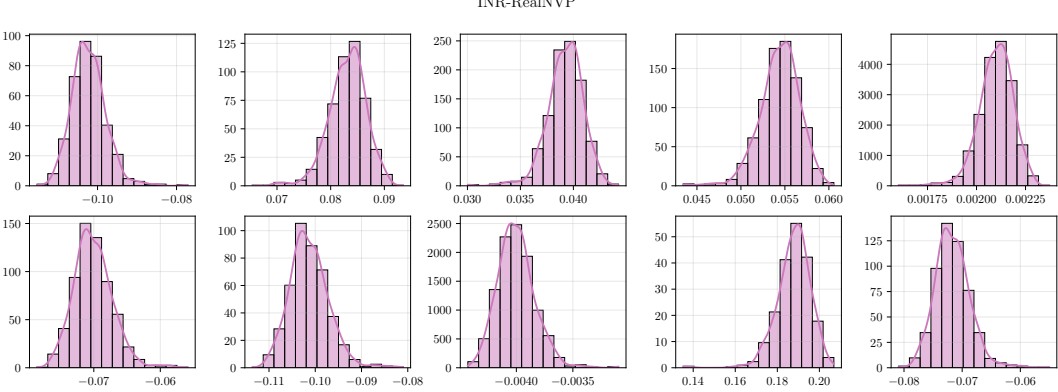

Figure 12: Empirical Covariance for the INR-RealNVP for the first linear layer of the regression network.

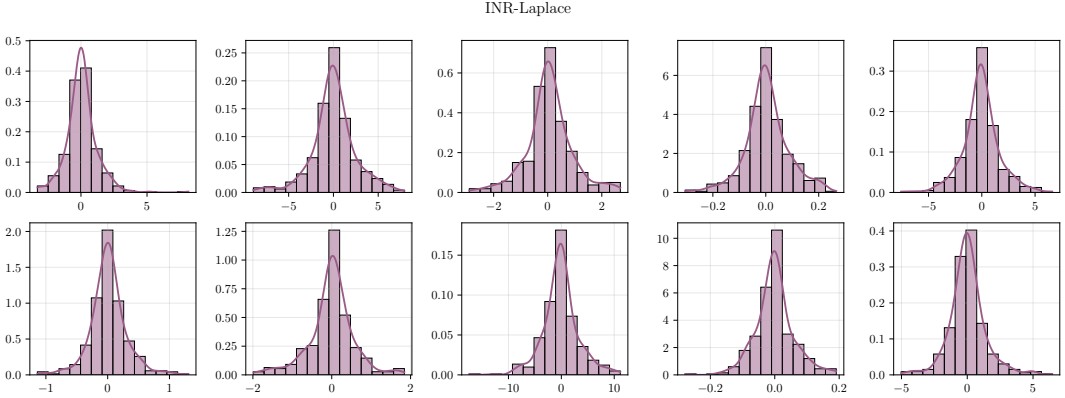

Figure 13: Empirical Covariance for the INR-Laplace for the first linear layer of the regression network.

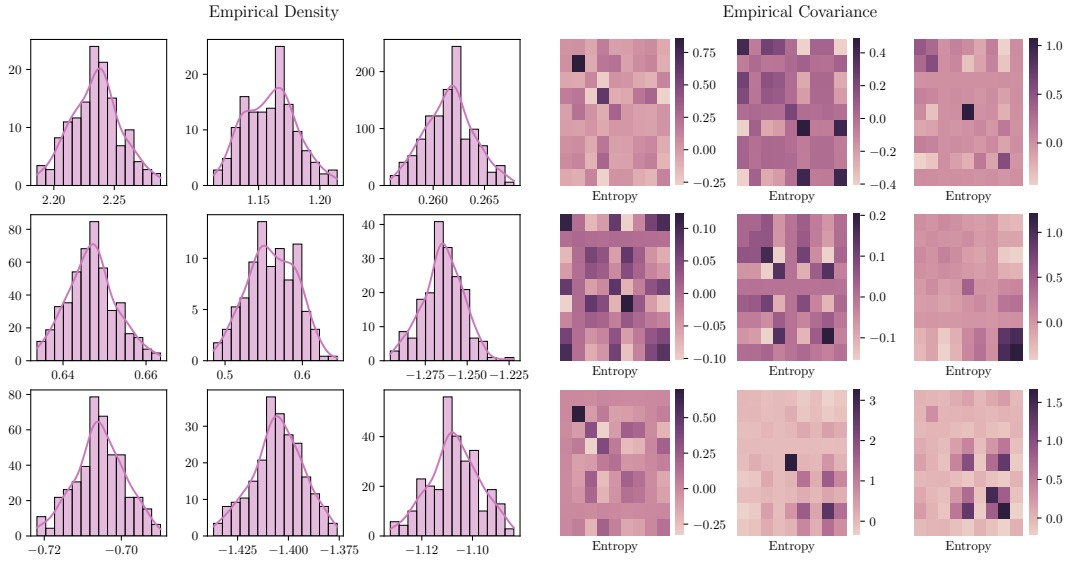

Figure 14: Empirical density histogram and empirical covariance for of kernel values of the first convolutional layer of ResNet-50 using INR-RealNVP.

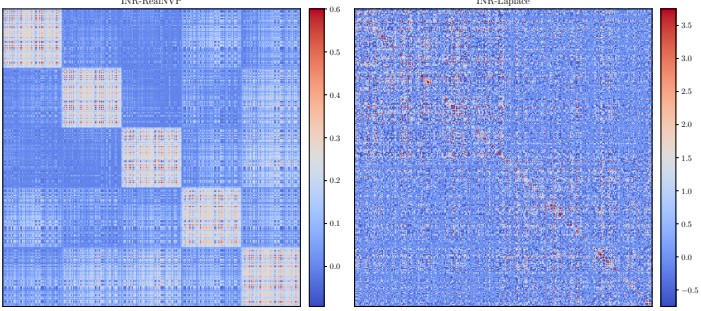

Figure 15: Empirical Covariance for the INR-RealNVP and INR-Laplace for the first linear layer of the regression network.

