# OpenReview forum: "Implicit Neural Representation Inference for Low-Dimensional Bayesian Deep Learning"
_ICLR.cc/2024/Conference — ICLR 2024 poster_

### Official Review · Reviewer_wVmX · 2023-10-27

**Soundness:** 3 good
**Presentation:** 4 excellent
**Contribution:** 3 good
**Rating:** 8
**Confidence:** 4

**Summary:**

The authors propose an approximate inference method for Bayesian neural networks. The idea is to introduce a second (auxiliary) network which computes an approximate posterior over the multiplicative noise applied to the deterministic weights of the main network. Such a multiplicative noise then induces an approximate predictive posterior distribution of the main network, they key object of interest in BNNs. The auxiliary network can be of much smaller size than the main one, and an approximate posterior over its weights can be approximated using one of the existing methods (e.g. Laplace, SWAG, Normalising Flows). The proposed method is shown to be competitive in comparison to a number of baselines.

**Strengths:**

+ An interesting approach to approximate BNN inference showing competitive performance
+ Clear presentation, the paper is easy to follow

**Weaknesses:**

The baselines used for comparison (Dropout, BbH, Ensembles) are relatively old methods (by Deep Learning standards of course). I'd be very interested to see how the proposed method compares to more modern approaches, e.g. those applying Laplace approximation directly on to weights of the main network.

A couple of minor points:
- Typo in the second to last line on page 3 (wf).
- Typo in Eq. (5) (w_{INR} should be in the subscript I guess?)

**Questions:**

- I wonder about the integer inputs (i.e. the tensor coordinates) to the INR network. Do you normalise these coordinates somehow or directly input the integers into the INR network? Did such an integer input space cause any problems during training?
- Why do you think that sinusoidal activations are particularly suitable for the INR network? Do you expect the result to deteriorate if you used other activations (e.g. sigmoid)?
- Why do you think the INR with 350 outperformed 4k and 10k versions on CIFAR in Fig. 1?
- I was very interested to see that the predictive uncertainty in Fig. 2 has a stationary structure (i.e. dependent on the distance from the observations) similar to a GP with a stationary kernel. The Dropout and Deep Ensembles baselines clearly don't have such a property, but I wonder how such a figure would look like if we used a Laplace approximation directly on some layers of the main network (without INR), e.g. similar to Kristiadi et al. (2020)? In other words, I wonder how specific this stationary uncertainty structure is to the inference using an auxiliary network?

---

> ### Author Response · Authors · 2023-11-17
>
> *”An interesting approach to approximate BNN inference showing competitive performance [..] Clear presentation, the paper is easy to follow”*
>
> Thank you, we appreciate your positive comments. We are very content that you found the presentation clear.

---

> ### Author Response · Authors · 2023-11-17
>
> *“The baselines used for comparison (Dropout, BbH, Ensembles) are relatively old methods (by Deep Learning standards of course). I'd be very interested to see how the proposed method compares to more modern approaches, e.g. those applying Laplace approximation directly on to weights of the main network.”*
>
> Following the reviewer’s recommendations we added an additional ablation study. We tried to measure the quality of proposed subspaces in terms of predictive uncertainty. Specifically we compare our INR low dimensional space with: Rank-1 (Dusenberry et al. 2020); Wasserstein subnetwork  (Daxberger et al. 2021) and partially stochastic ResNets from (Sharma, Mrinank, et al. partially).
> We trained (each method) combined with a Resnet18 for 100 epochs in both Cifar10 and Cifar100 datasets while keeping the approximate inference method fixed same across all low dimensional spaces.
>
> * Sharma, Mrinank, et al. "Do Bayesian Neural Networks Need To Be Fully Stochastic?." International Conference on Artificial Intelligence and Statistics. PMLR, 2023.
>
>  **RANK1**
>
>  SWAG - CIFAR10 In-Dist Test data Accuracy: 91.78 LL: -0.4187 Error: 0.082 Brier: 0.1343 ECE: 0.0522
>
>  SWAG - CIFAR10 Corrupted Test data Accuracy: 77.80 LL: -1.2537 Error: 0.222 Brier: 0.3596 ECE: 0.1469
>
>  Laplace - CIFAR10 In-Dist Test data Accuracy: 91.01 LL: -1.5630 Error: 0.090 Brier: 0.6872 ECE: 0.6871
>
>  Laplace - CIFAR10 Corrupted Test data Accuracy: 78.07 LL: -1.7068 Error: 0.220 Brier: 0.7396 ECE: 0.5737
>
> -----------------------------------------------------------------------------------------------------------------------------------------------------
>
> SWAG -   CIFAR100 In-Dist Test data         Accuracy: 65.84 LL: -2.29 Error: 0.34 Brier: 0.55 ECE: 0.228
>
> SWAG -   CIFAR100 Corrupted Test data   Accuracy: 42.80 LL: -4.77 Error: 0.57 Brier: 0.92 ECE: 0.398
>
>  Laplace - CIFAR100 In-Dist Test data        Accuracy: 69.00 LL: -4.01 Error: 0.31 Brier:  0.9714 ECE: 0.668
>
>  Laplace - CIFAR100 Corrupted Test data   Accuracy: 42.00 LL: -4.25 Error: 0.58 Brier 0.979 ECE: 0.401
>
>  **INR**
>
>  SWAG -   CIFAR10 In-Dist Test data Accuracy: 92.17 LL: -0.384 Error: 0.078 Brier: 0.1296 ECE: 0.0498
>
>  SWAG -   CIFAR10 Corrupted Test data Accuracy: 78.60 LL: -1.164 Error: 0.214 Brier: 0.3566 ECE: 0.1447
>
>  Laplace - CIFAR10 In-Dist Test data Accuracy: 89.0 LL: -1.563 Error: 0.110 Brier:  0.6869 ECE: 0.6647
>
>  Laplace - CIFAR10 Corrupted Test data Accuracy: 81.0 LL: -1.660 Error: 0.190 Brier: 0.7156 ECE: 0.5890
>
> ------------------------------------------------------------------------------------------------------------------------------------------------------
>
>  SWAG -   CIFAR100 In-Dist Test data Accuracy: 69.12 LL: -2.094 Error: 0.308 Brier: 0.5006 ECE: 0.2046
>
>  SWAG -   CIFAR100 Corrupted Test data Accuracy: 46.5 LL: -4.1878 Error: 0.535 Brier: 0.8418 ECE: 0.3640
>
>  Laplace - CIFAR100 In-Dist Test data Accuracy: 70.0 LL: -3.9172 Error: 0.300 Brier:  0.967 ECE: 0.6747
>
> Laplace - CIFAR100 Corrupted Test data   Accuracy: 42.0 LL: -4.199 Error: 0.58 Brier:0.9770 ECE: 0.396
>
>  **SUBNETWORK**
>
>  SWAG -   CIFAR10 In-Dist Test data Accuracy: 92.54 LL: -0.4251 Error: 0.074 Brier: 0.1255 ECE: 0.0491
>
>  SWAG -   CIFAR10 Corrupted Test data Accuracy: 76.90 LL: -1.4599 Error: 0.231 Brier: 0.3845 ECE: 0.1732
>
>  Laplace - CIFAR10 In-Dist Test data Accuracy: 91.0 LL: -1.551723 Error: 0.090 Brier: 0.682 ECE: 0.6823
>
>  Laplace - CIFAR10 Corrupted Test data  Accuracy:  81.0 LL: -1.650211 Error: 0.190 Brier: 0.7134 ECE: 0.5886
>
> ------------------------------------------------------------------------------------------------------------------------------------------------------
>
>  SWAG -   CIFAR100 In-Dist Test data Accuracy: 69.86 LL: -2.1430 Error: 0.30 Brier: 0.49 ECE: 0.207
>
>  SWAG -   CIFAR100 Corrupted Test data Accuracy: 49.0 LL: -3.9721 Error: 0.51 Brier: 0.82 ECE: 0.3483
>
>  Laplace - CIFAR100 In-Dist Test data Accuracy: 68.0 LL: -3.9505 Error: 0.32 Brier: 0.9682 ECE: 0.655
>
>  Laplace - CIFAR100 Corrupted Test data Accuracy: 49.0 LL: -4.1309 Error: 0.51 Brier: 0.974 ECE: 0.466
>
>  **PARTIALLY STOCHASTIC**
>
>  SWAG -   CIFAR10 In-Dist Test data Accuracy: 92.54 LL: -0.4251 Error: 0.074 Brier: 0.1255 ECE: 0.0491
>
>  SWAG -   CIFAR10 Corrupted Test data Accuracy: 76.92 LL: -1.4499 Error: 0.201 Brier: 0.3806 ECE: 0.1730
>
>  Laplace - CIFAR10 In-Dist Test data Accuracy: 90.8 LL: -1.561 Error: 0.091 Brier: 0.68 ECE: 0.702
>
>  Laplace - CIFAR10 Corrupted Test data Accuracy: 80.0 LL: -1.67 Error: 0.21 Brier: 0.72 ECE: 0.59
>
>
> ***thread continued.........***

---

> ### Author Response · Authors · 2023-11-17
>
> ----------------------------------------------------------------------------------------------------------------------------------------------------
>
>  SWAG - CIFAR100 In-Dist Test data Accuracy: 69.85 LL: -2.1430 Error: 0.3015 Brier: 0.4997 ECE: 0.2078
>
>  SWAG - CIFAR100 Corrupted Test data Accuracy: 49.0 LL: -3.9721 Error: 0.51 Brier: 0.8218 ECE: 0.3483
>
>  Laplace - CIFAR100 In-Dist Test data Accuracy: 66.0 LL: -3.9903 Error: 0.34 Brier:  0.978 ECE: 0.6377
>
>  Laplace - CIFAR100 Corrupted Test data Accuracy: 49.0 LL: -4.1815 Error: 0.51 Brier 0.989 ECE: 0.4709
>
> \*For SWAG and Linearized Laplace with GGN, in order to be able to run across low dimensional spaces we choose the covariance to have Diagonal structure.
>
> As for the results, we believe that in both datasets there is at the very least a trend in favor of both proposed INR-x methods. The results validate to a considerable degree the premise of the proposed methods: Instead of choosing a subset or subnet following the rationale of the corresponding methods, the INR produces "ξ" outputs that endow the full network with the desirable stochasticity, while keeping the dimensionality of the random process that we want to do inference upon at a low level.

---

> ### Author Response · Authors · 2023-11-17
>
> *“I wonder about the integer inputs (i.e. the tensor coordinates) to the INR network. Do you normalize these coordinates somehow or directly input the integers into the INR network? Did such an integer input space cause any problems during training?”*
>
> Following the SIREN paper Sitzmann et al. (2020). The tensor coordinate inputs of the hypernetwork are real numbers normalized to [-1,1]. All the technical details for each experiment are in the Appendix but we could add a separate paragraph dedicated to the INR technical details.

---

> ### Author Response · Authors · 2023-11-17
>
> *“Why do you think that sinusoidal activations are particularly suitable for the INR network? Do you expect the result to deteriorate if you used other activations (e.g. sigmoid)?”*
>
> We trained Resnet18 in both CIFAR10 and CIFAR100 for 100 epochs to evaluate the predictive capabilities of the Sinusoidal hyper-network.
> Here we have a comparison of Sine vs ReLU. If you insist on including sigmoid or another activation function specifically, we'll be happy to include a comparison.
>
>  **CIFAR10**
>
>  RELU_MAP Accuracy: 91.11 LL: -0.4891 Error: 0.088 Brier: 0.1484 ECE: 0.05891
>
>  SINE MAP   Accuracy: 91.70 LL: -0.4449 Error: 0.083 Brier: 0.138 ECE: 0.05401
>
>  **CIFAR100**
>
>  RELU_MAP Accuracy: 67.79 LL: -2.544 Error: 0.3221 Brier: 0.537 ECE: 0.23232
>
>  SINE MAP   Accuracy: 68.49 LL: -2.3990 Error: 0.3151 Brier: 0.527 ECE: 0.2256
>
> We find that Sine/Periodic activations – the “default” choice in SIREN – slightly outperforms a hypernet with ReLU activations. Still, results are very close, though there is a trend in favor of sine in all benchmarks.
> The original motivation behind using the sine activation is related to modeling high-frequency content, which translates as details in structured signals such as images or video [Sitzmann 2020]. We can however see this “in the top of its head”, so to speak: in structured signals we care *more* for low-frequency content, and high-frequency is a “good-to-have” content. We can interpret an input semantically if we see its low frequencies, but not necessarily vice versa. For example, image compression will invariably throw away high frequencies first, and the last frequencies to lose will be the lower ones.
> Our conjecture is as follows: When using an INR to model *perturbations*, we are faced with a different situation, that corresponds to a different “frequency landscape” (perhaps even different than the one of model *weights*). In particular, we do not have any reason to differentiate lower or higher frequency content in any respect. We “care” for all frequencies, so we need to have a good way to model high frequencies as well. Perhaps this is the reason the sine activation gives a small edge over ReLU.

---

> ### Author Response · Authors · 2023-11-17
>
> *“Why do you think the INR with 350 outperformed 4k and 10k versions on CIFAR in Fig. 1?”*
>
> We believe that the reason is related to the complexity of each problem. In CIFAR, a smaller model seems to be “enough” in terms of capacity, while the bigger versions are overly complex. Note that in Corrupted CIFAR (columns on the right) the situation is reversed, because the problem is comparatively more difficult, and we need the extra model capacity.

---

> ### Author Response · Authors · 2023-11-17
>
> *“I was very interested to see that the predictive uncertainty in Fig. 2 has a stationary structure (i.e. dependent on the distance from the observations) similar to a GP with a stationary kernel. The Dropout and Deep Ensembles baselines clearly don't have such a property, but I wonder how such a figure would look like if we used a Laplace approximation directly on some layers of the main network (without INR), e.g. similar to Kristiadi et al. (2020)? In other words, I wonder how specific this stationary uncertainty structure is to the inference using an auxiliary network?”*
>
> The stationary structure (or in-between-uncertainty) is one of the benefits of the Linearized Laplace approximation as shown in multiple works (Kristiadi et al. (2020), Daxberger et al. (2021), Immer et al., 2021), ). In this Fig. 2 what we want to highlight is that the low dimensional INR space we propose is able to maintain the appealing characteristics of the approximate inference methods applied (in this particular case the stationary structure of the Lin. Laplace )

---

> > ### Comment · Reviewer_wVmX · 2023-11-20
> > **Thank you for the reply!**
> >
> > Thank you for your detailed reply. I don't have further questions at this stage and I confirm my positive view on this submission.

---

> > > ### Author Response · Authors · 2023-11-22
> > > **Thank you!**
> > >
> > > Thank you for the effort you put for the review, and thank you for appreciating our work. We will do our best to improve the final manuscript as per your comments.

---

### Official Review · Reviewer_Ji8X · 2023-10-30

**Soundness:** 2 fair
**Presentation:** 2 fair
**Contribution:** 2 fair
**Rating:** 5
**Confidence:** 4

**Summary:**

This work proposes so-called implicit neural representation inference for Bayesian Neural Networks. In the past, several "subspace" inference frameworks have been devised, where the aim is to only model smaller part of the weight space in a Bayesian manner. This type of approaches promise to better scale the approximate Bayesian inference for deep learning, while making the inference procedure more accurate. Building upon, this paper proposes to obtain the "subspace" of weights using implicit neural representation. The paper shows how the proposed method can enhance popular frameworks such as Laplace Approximation, SWAG and normalizing flows. Experiments are conducted on UCI, cifar10 and cifar100 and the approach is compared against the baselines with performs inference over all the parameters of neural networks, as oppose to the subspace.

**Strengths:**

In my view, the paper has the following strengths:

- the idea of using implicit neural representation for improving Bayesian Neural Network is interesting and novel.

- Implicit neural representations are currently a popular topic, and may be therefore relevant to many researchers.

**Weaknesses:**

On the other-hand, I think the paper has several rooms to improve for a publication.

- The paper could improve in terms of its clarity.

Specifically, I find it difficult to parse section 3.1. I get the problem statement, but the parts on implicit neural representation with the SIREN model was difficult to understand. It would help to have a figure on this since there are many notations introduced. The text contains many mathematical symbols, which would have been explained differently.

SIREN should be explained more in depth since I think it is an important technical detail. Also, some technical details on how implicit neural representation is obtained, and the general working principles, like an algorithm behind, would be helpful for the reader.

- Experiments may become more solid with other choices of baselines and datasets.

First, there has been many subspace approaches and also many approaches that attempts to sparsely represent model uncertainty. Some examples are references in the paper: Kristiadi, Dusenberry, Daxberger, etc. I think the experiments should compare to these baselines as well, which can really show the advantages of implicit neural representation over existing methods within the same class of approaches. Comparisons to full weight space seem not natural.

Moreover, the paper uses UCI and CIFAR as main datasets. I would have liked the paper more if "uncertainty baselines" from google was used, as such works represent the more upto date standard in experimental protocol. Speaking of the protocol, the included baselines seem not very consistent, e.g., Figure 1 misses laplace approximation, figure 2 again selectively reports INR Laplace and contains no SWAG, figure 3 misses swag, INR swag, etc.

Overall, I recommend a weak rejection. Improving the technical quality and clarity of the presentation would be meaningful here.

**Questions:**

1.In line with section 3.1, is it possible to explain why INR is advantages to learn the subspace?  I could not get why it might be a good idea.

2. Another question is on expressiveness Vs accuracy of the Bayesian inference. Basically, the weight space is very large. Having a simple distribution in such a complex high dimensional space can be already advantageous in terms of expressiveness of the probability distribution. Of course, a natural direction has been also improving the expressiveness through structured distribution, correlations amongst layers, etc. On the other hand, what these subspace approaches do is to model with smaller part of the network, which can actually reduce the overall expressiveness of the distribution, though overall inference might be easier and accurate. Then the question is, when should we look into the approaches for expressiveness, and when should we look into the subspace approaches?

3. Why not only take last few layers of the network and make them probabilistic? What are advantageous of using INR against very simple baselines as taking last one or three layers? I think it might be interesting to include them as a baselines in the experiments.

---

> ### Author Response · Authors · 2023-11-17
>
> *“The paper could improve in terms of its clarity. [...] Specifically, I find it difficult to parse section 3.1. I get the problem statement, but the parts on implicit neural representation with the SIREN model was difficult to understand. It would help to have a figure on this since there are many notations introduced. The text contains many mathematical symbols, which would have been explained differently.”*
>
> Thank you for your comment. We will do our best to clarify this section in the final text. In the meanwhile, we’d be happy to explain any step or detail of the method you would like to. We will add a figure representing the proposed model.
>
> We added a graphical assessment which depicts our method in simple MLP Training Setting, where the main network consists of 4 linear layers and the INR hypernetwork has 2 layers producing accordingly 4 sets of ξ factors (see https://freeimage.host/i/JnnQtyb).

---

> ### Author Response · Authors · 2023-11-17
>
> *“SIREN should be explained more in depth since I think it is an important technical detail. Also, some technical details on how implicit neural representation is obtained, and the general working principles, like an algorithm behind, would be helpful for the reader.”*
>
> We gladly add high level pseudocode to introduce our methods behavior in training and inference settings:
>
> **Algorithm 1** Training
> ```
>  Inputs: I (indices of main network weights), Net (main network), INR (INR hypernetwork), Dataset
>  for number of epochs do
>  	for x,y in Dataset do
>  		ξ = INR(I)
>  		y* = Net(x,ξ)
>  		loss = (y,y*)
>  		update INR w.r.t loss
>  		update Net w.r.t loss
>  	end
>  end
> ```
>
> **Algorithm 2** Inference
> ```
>  Inputs: I (indices of main network weights), Net (main network), INR (INR hypernetwork), Test_set
>  Ap.-In (Approximate inference method*) MC_samples (Number of Monte Carlo samples)
>  for x in Test_set do
>  	for MC_samples do
>  		ξj = Ap.-In(INR, I)
> 		y* = Net(x,ξj)
>  	end
>  	calculate y* statistics
>  end
> ```
>
> (In this setting a post training Monte Carlo-based approximate inference method is implied).

---

> ### Author Response · Authors · 2023-11-17
>
> *"I would have liked the paper more if "uncertainty baselines" from google was used, as such works represent the more upto date standard in experimental protocol. Speaking of the protocol, the included baselines seem not very consistent, e.g., Figure 1 misses laplace approximation, figure 2 again selectively reports INR Laplace and contains no SWAG, figure 3 misses swag, INR swag, etc.”*
>
> We agree that “Uncertainty baselines” is definitely useful, and we have tried to include as many benchmarks that appear in it.
>
> In our work we tried to validate our INR-’space’ combined with a variety of approximate inference methods. Both in classification and regression experiments all the three posterior approximations combined our method yield good and calibrated results.
> There are some reasons why some methods do not appear in some figures. For example in Fig1 we wanted to also measure the diversity so we wanted a Monte Carlo based method and to save space we chose SWAG. Also in Fig2 we wanted to highlight the benefits of Laplace approximation in particular. In general we followed a pattern that in Regression we use (Laplace/RealNVP) , and in Classification (SWAG/RealNVP). We will gladly include all three methods in all experiments in the appendix.

---

> ### Author Response · Authors · 2023-11-17
>
> *"Experiments may become more solid with other choices of baselines and datasets. First, there has been many subspace approaches and also many approaches that attempts to sparsely represent model uncertainty."*
>
> For our UCI regression experiments we added another strong baseline. For the small MLP network we use we were able to compute the full GGN matrix in the Laplace approximation of the main network. We copy the result in the form a figure (https://freeimage.host/i/Jnn6z6g).

---

> ### Author Response · Authors · 2023-11-17
>
> *“1.In line with section 3.1, is it possible to explain why INR is advantageous to learn the subspace? I could not get why it might be a good idea.”*
>
> Our inspiration comes from works where Implicit Neural Representation are used to learn sets of network weights. For example in Romero et al. (2021a), convolutional kernels are represented in terms of INR-based Multiplicative Anisotropic Gabor Networks. Another recent example is Ashkenazi, Maor, et al. "NeRN: Learning Neural Representations for Neural Networks." from ICLR 2022.

---

> ### Author Response · Authors · 2023-11-17
>
> *“Why not only take last few layers of the network and make them probabilistic? What are the advantages of using INR against very simple baselines as taking last one or three layers? I think it might be interesting to include them as a baselines in the experiments.”*
>
> Following the reviewer’s recommendations we added an additional ablation study. We tried to measure the quality of proposed subspaces in terms of predictive uncertainty. Specifically we compare our INR low dimensional space with: Rank-1 (Dusenberry et al. 2020); Wasserstein subnetwork  (Daxberger et al. 2021) and partially stochastic ResNets from (Sharma, Mrinank, et al. partially).
> We trained (each method) combined with a Resnet18 for 100 epochs in both Cifar10 and Cifar100 datasets while keeping the approximate inference method fixed same across all low dimensional spaces.
>
> * Sharma, Mrinank, et al. "Do Bayesian Neural Networks Need To Be Fully Stochastic?." International Conference on Artificial Intelligence and Statistics. PMLR, 2023.
>
>  **RANK1**
>
>  SWAG - CIFAR10 In-Dist Test data Accuracy: 91.78 LL: -0.4187 Error: 0.082 Brier: 0.1343 ECE: 0.0522
>
>  SWAG - CIFAR10 Corrupted Test data Accuracy: 77.80 LL: -1.2537 Error: 0.222 Brier: 0.3596 ECE: 0.1469
>
>  Laplace - CIFAR10 In-Dist Test data Accuracy: 91.01 LL: -1.5630 Error: 0.090 Brier: 0.6872 ECE: 0.6871
>
>  Laplace - CIFAR10 Corrupted Test data Accuracy: 78.07 LL: -1.7068 Error: 0.220 Brier: 0.7396 ECE: 0.5737
>
> -----------------------------------------------------------------------------------------------------------------------------------------------------
>
> SWAG -   CIFAR100 In-Dist Test data         Accuracy: 65.84 LL: -2.29 Error: 0.34 Brier: 0.55 ECE: 0.228
>
> SWAG -   CIFAR100 Corrupted Test data   Accuracy: 42.80 LL: -4.77 Error: 0.57 Brier: 0.92 ECE: 0.398
>
>  Laplace - CIFAR100 In-Dist Test data        Accuracy: 69.00 LL: -4.01 Error: 0.31 Brier:  0.9714 ECE: 0.668
>
>  Laplace - CIFAR100 Corrupted Test data   Accuracy: 42.00 LL: -4.25 Error: 0.58 Brier 0.979 ECE: 0.401
>
>  **INR**
>
>  SWAG -   CIFAR10 In-Dist Test data Accuracy: 92.17 LL: -0.384 Error: 0.078 Brier: 0.1296 ECE: 0.0498
>
>  SWAG -   CIFAR10 Corrupted Test data Accuracy: 78.60 LL: -1.164 Error: 0.214 Brier: 0.3566 ECE: 0.1447
>
>  Laplace - CIFAR10 In-Dist Test data Accuracy: 89.0 LL: -1.563 Error: 0.110 Brier:  0.6869 ECE: 0.6647
>
>  Laplace - CIFAR10 Corrupted Test data Accuracy: 81.0 LL: -1.660 Error: 0.190 Brier: 0.7156 ECE: 0.5890
>
> ------------------------------------------------------------------------------------------------------------------------------------------------------
>
>  SWAG -   CIFAR100 In-Dist Test data Accuracy: 69.12 LL: -2.094 Error: 0.308 Brier: 0.5006 ECE: 0.2046
>
>  SWAG -   CIFAR100 Corrupted Test data Accuracy: 46.5 LL: -4.1878 Error: 0.535 Brier: 0.8418 ECE: 0.3640
>
>  Laplace - CIFAR100 In-Dist Test data Accuracy: 70.0 LL: -3.9172 Error: 0.300 Brier:  0.967 ECE: 0.6747
>
> Laplace - CIFAR100 Corrupted Test data   Accuracy: 42.0 LL: -4.199 Error: 0.58 Brier:0.9770 ECE: 0.396
>
>  **SUBNETWORK**
>
>  SWAG -   CIFAR10 In-Dist Test data Accuracy: 92.54 LL: -0.4251 Error: 0.074 Brier: 0.1255 ECE: 0.0491
>
>  SWAG -   CIFAR10 Corrupted Test data Accuracy: 76.90 LL: -1.4599 Error: 0.231 Brier: 0.3845 ECE: 0.1732
>
>  Laplace - CIFAR10 In-Dist Test data Accuracy: 91.0 LL: -1.551723 Error: 0.090 Brier: 0.682 ECE: 0.6823
>
>  Laplace - CIFAR10 Corrupted Test data  Accuracy:  81.0 LL: -1.650211 Error: 0.190 Brier: 0.7134 ECE: 0.5886
>
> ------------------------------------------------------------------------------------------------------------------------------------------------------
>
>  SWAG -   CIFAR100 In-Dist Test data Accuracy: 69.86 LL: -2.1430 Error: 0.30 Brier: 0.49 ECE: 0.207
>
>  SWAG -   CIFAR100 Corrupted Test data Accuracy: 49.0 LL: -3.9721 Error: 0.51 Brier: 0.82 ECE: 0.3483
>
>  Laplace - CIFAR100 In-Dist Test data Accuracy: 68.0 LL: -3.9505 Error: 0.32 Brier: 0.9682 ECE: 0.655
>
>  Laplace - CIFAR100 Corrupted Test data Accuracy: 49.0 LL: -4.1309 Error: 0.51 Brier: 0.974 ECE: 0.466
>
>  **PARTIALLY STOCHASTIC**
>
>  SWAG -   CIFAR10 In-Dist Test data Accuracy: 92.54 LL: -0.4251 Error: 0.074 Brier: 0.1255 ECE: 0.0491
>
>  SWAG -   CIFAR10 Corrupted Test data Accuracy: 76.92 LL: -1.4499 Error: 0.201 Brier: 0.3806 ECE: 0.1730
>
>  Laplace - CIFAR10 In-Dist Test data Accuracy: 90.8 LL: -1.561 Error: 0.091 Brier: 0.68 ECE: 0.702
>
>  Laplace - CIFAR10 Corrupted Test data Accuracy: 80.0 LL: -1.67 Error: 0.21 Brier: 0.72 ECE: 0.59
>
>
> ***thread continued.........***

---

> ### Author Response · Authors · 2023-11-17
>
> ----------------------------------------------------------------------------------------------------------------------------------------------------
>
>  SWAG - CIFAR100 In-Dist Test data Accuracy: 69.85 LL: -2.1430 Error: 0.3015 Brier: 0.4997 ECE: 0.2078
>
>  SWAG - CIFAR100 Corrupted Test data Accuracy: 49.0 LL: -3.9721 Error: 0.51 Brier: 0.8218 ECE: 0.3483
>
>  Laplace - CIFAR100 In-Dist Test data Accuracy: 66.0 LL: -3.9903 Error: 0.34 Brier:  0.978 ECE: 0.6377
>
>  Laplace - CIFAR100 Corrupted Test data Accuracy: 49.0 LL: -4.1815 Error: 0.51 Brier 0.989 ECE: 0.4709
>
> \*For SWAG and Linearized Laplace with GGN, in order to be able to run across low dimensional spaces we choose the covariance to have Diagonal structure.
>
> As for the results, we believe that in both datasets there is at the very least a trend in favor of both proposed INR-x methods. The results validate to a considerable degree the premise of the proposed methods: Instead of choosing a subset or subnet following the rationale of the corresponding methods, the INR produces "ξ" outputs that endow the full network with the desirable stochasticity, while keeping the dimensionality of the random process that we want to do inference upon at a low level.

---

> ### Author Response · Authors · 2023-11-18
>
> *“Another question is on expressiveness Vs accuracy of the Bayesian inference. Basically, the weight space is very large. Having a simple distribution in such a complex high dimensional space can be already advantageous in terms of expressiveness of the probability distribution. Of course, a natural direction has been also improving the expressiveness through structured distribution, correlations amongst layers, etc. On the other hand, what these subspace approaches do is to model with smaller part of the network, which can actually reduce the overall expressiveness of the distribution, though overall inference might be easier and accurate. Then the question is, when should we look into the approaches for expressiveness, and when should we look into the subspace approaches?”*
>
> This is an interesting question.
>
> Our take is that there are multiple trade-offs behind choices that we have to make in the context of a model solved with Bayesian inference. A more complex model should correspond to a more complex weight space, which translates to a more difficult optimization problem. Therefore – if we understand your position correctly, we have a decision to make regarding our “complexity budget”, so to speak. So there is one issue concerning the problem of whether we should prioritize learning an accurate estimator, or ensuring that we can have a learning system that will output a calibrated measure of uncertainty. In the context of the proposed INR-based approach, this is related to the complexity, expressiveness and capacity of the main network and the (SIREN) hypernetwork respectively.
>
> Perhaps closer to your point of view is the question regarding how to proceed more efficiently with respect to choices regarding inference. If we have to work on the entire space of weights as the domain for our posterior distribution, in *practice* we need to make concessions in inference, which translates as working with uncorrelated estimates per weight, or correlations only on the level of layer, or KFAC, and so on. The other approach, closer to what we do in the current work, is to choose some lower-dimensionality space to work with a minimum of inference constraints. This is what we do in the current work, and we think that a comparatively small INR is a very good way to proceed here. Concerning “when” is one way to proceed better than the other, our results indicate that the INR-based approach comes with clear advantages. We will note though, that this is related to the complexity of the problems we try to solve. We tried our best to include as many baselines and ablations in this respect.
>
> As the future of the probabilistic ML subfield moves towards larger and more complex baselines, time will tell how the “expressiveness vs accuracy” tradeoff will shape new models and approaches.

---

### Official Review · Reviewer_VEKs · 2023-10-31

**Soundness:** 2 fair
**Presentation:** 2 fair
**Contribution:** 3 good
**Rating:** 5
**Confidence:** 3

**Summary:**

The authors present a new framework for approximate Bayesian inference in neural networks based on low-dimensional hypernetwork representations of weight perturbations. In this framework, hypernetworks take in weight coordinates as input and output perturbation factors for each weight. An approximate posterior is fitted for the hypernetwork and perturbations are repeatedly sampled and multiplied with the main network weights to perform Bayesian model averaging. Three alternative Bayesian hypernetworks are considered: sinusoidal representation networks (SIRENs) with Laplace and SWA-Gaussian approximate posteriors, and normalizing flows with Real Non-Volume Preserving (RealNVP) transformations. The posteriors are benchmarked on UCI regression and gap datasets, and out-of-distribution (OOD) detection is tested on CIFAR-10 image recognition.

**Strengths:**

- The proposed framework introduces a novel combination of approximate Bayesian inference and implicit neural representations as hypernetworks which, to the best of my knowledge, has not been explored in prior research.
- The paper offers valuable insights, e.g. in considering the benefit of multiplicative over additive perturbances, and in finding that CNN models benefit from shared representations.
- The evaluation of out-of-distribution (OOD) detection on the CIFAR-10 dataset is thorough and detailed. This plays a large role in showing the practical utility of the proposed approach.

**Weaknesses:**

Regarding the method:
- The derivation of the Laplace approximation is confusing: In Equations (4) and (5) a Laplace approximation with full Hessian is derived. While a full Hessian for the hypernetwork weights may be tractable for small enough networks, this approximation does not correspond to the closed form posterior of Equation (7). This closed form corresponds to linearized Laplace inference with generalized Gauss-Newton (GGN) approximation to the Hessian. It is unclear if the model evaluated in the experiments uses a full Hessian or the GGN approximation with closed form.
- The use of SIREN activation suggests that the hypernetworks need to model high frequency representations of the weight perturbations. The SIREN models presented in [Sitzmann et al., 2020] benefit from consistency and repeating patterns in the signals they are fitting, making interpolation easier. There might be some form of structural consistency in neighboring weight perturbations for CNN kernels, but it seems unlikely for the weights of linear layers. I also expect that RealNVP hypernetworks have a harder time fitting high frequency representations when comparing to SIREN. This matter is only very briefly brought up in the discussion of Figures 9 and 10.

Regarding experiments:
- The methods are benchmarked against a last-layer Laplace approximation, however a block-diagonal Kronecker-factorized (KFAC) Laplace approximation should also be considered since this corresponds better to a full network variant in the linearized Laplace family of approximations.
- Both theoretical and experimental runtimes and memory requirements are not discussed. It is unclear if the proposed models take significantly more time for training or inference, since in theory a forward pass of the hypernetwork is required for every "main" network weight.

**Questions:**

- Does the INR-Laplace model in the experiments employ a full Hessian or a GGN approximation? Do you make any additional approximations (e.g. Kronecker-factorization) ?
- Can you further evaluate the effect of INR network size on model performance and quality of the uncertainty estimates? Figure 1 already does this for CIFAR-10 but it would be interesting to see for MLPs on UCI and UCI-gap datasets, with INR-Laplace and INR-RealNVP, and perhaps on wider sets of network sizes. Would also be helpful to include error bars for these figures.
- Have you tried using ReLU hypernetworks? Do these fail to recover high frequency representations of the perturbations compared to SIREN?
- Do you have an intuition for why multiplicative perturbations are so successful compared to additive perturbations of the weights? Perhaps uncertainty is improved when perturbations are scaled up by the magnitude of the weight?
- Is there a reason for INR-SWAG appearing in some experiments and INR-Laplace in others?

---

> ### Author Response · Authors · 2023-11-17
>
> *“The derivation of the Laplace approximation is confusing: In Equations (4) and (5) a Laplace approximation with full Hessian is derived. While a full Hessian for the hypernetwork weights may be tractable for small enough networks, this approximation does not correspond to the closed form posterior of Equation (7). This closed form corresponds to linearized Laplace inference with generalized Gauss-Newton (GGN) approximation to the Hessian. It is unclear if the model evaluated in the experiments uses a full Hessian or the GGN approximation with closed form. [... Question 1:] Does the INR-Laplace model in the experiments employ a full Hessian or a GGN approximation? Do you make any additional approximations (e.g. Kronecker-factorization) ?”*
>
>  Indeed, in practice we do not use the full Hessian here. We use linearized Laplace with GGN approximation to the Hessian. Eq.4 and 5 correspond to the more general case, which is nevertheless consistent with the rest of the model in theory. We do not do KFAC or other approximations concerning the Hessian.

---

> ### Author Response · Authors · 2023-11-17
>
> *“The use of SIREN activation suggests that the hypernetworks need to model high frequency representations of the weight perturbations. The SIREN models presented in [Sitzmann et al., 2020] benefit from consistency and repeating patterns in the signals they are fitting, making interpolation easier. There might be some form of structural consistency in neighboring weight perturbations for CNN kernels, but it seems unlikely for the weights of linear layers. [..] This matter is only very briefly brought up in the discussion of Figures 9 and 10. [..] Have you tried using ReLU hypernetworks? Do these fail to recover high frequency representations of the perturbations compared to SIREN?”*
>
> We trained Resnet18 in both CIFAR10 and CIFAR100 for 100 epochs to evaluate the predictive capabilities of the Sinusoidal hyper-network:
>
>  **CIFAR10**
>
>  RELU_MAP Accuracy: 91.11 LL: -0.4891 Error: 0.088 Brier: 0.1484 ECE: 0.05891
>
>  SINE MAP   Accuracy: 91.70 LL: -0.4449 Error: 0.083 Brier: 0.138 ECE: 0.05401
>
>  **CIFAR100**
>
>  RELU_MAP Accuracy: 67.79 LL: -2.544 Error: 0.3221 Brier: 0.537 ECE: 0.23232
>
>  SINE MAP   Accuracy: 68.49 LL: -2.3990 Error: 0.3151 Brier: 0.527 ECE: 0.2256
>
> We find that Sine/Periodic activations – the “default” choice in SIREN – slightly outperforms a hypernet with ReLU activations. Still, results are very close, though there is a trend in favor of sine in all benchmarks.
> The original motivation behind using the sine activation is related to modeling high-frequency content, which translates as details in structured signals such as images or video [Sitzmann 2020]. We can however see this “in the top of its head”, so to speak: in structured signals we care *more* for low-frequency content, and high-frequency is a “good-to-have” content. We can interpret an input semantically if we see its low frequencies, but not necessarily vice versa. For example, image compression will invariably throw away high frequencies first, and the last frequencies to lose will be the lower ones.
> Our conjecture is as follows: When using an INR to model *perturbations*, we are faced with a different situation, that corresponds to a different “frequency landscape” (perhaps even different than the one of model *weights*). In particular, we do not have any reason to differentiate lower or higher frequency content in any respect. We “care” for all frequencies, so we need to have a good way to model high frequencies as well. Perhaps this is the reason the sine activation gives a small edge over ReLU.
>
> *“I also expect that RealNVP hypernetworks have a harder time fitting high frequency representations when compared to SIREN.”*
>
> We use RealNVP as a way to model the posterior approximation q(w_INR). SIREN is the architecture of the hypernetwork, so we can’t really compare the two.

---

> ### Author Response · Authors · 2023-11-17
>
> *“The methods are benchmarked against a last-layer Laplace approximation, however a block-diagonal Kronecker-factorized (KFAC) Laplace approximation should also be considered since this corresponds better to a full network variant in the linearized Laplace family of approximations.”*
>
> Thank you for the suggestion. We agree that this would be a very useful experiment. However, we find it very difficult to complete any meaningful set of experiments on KFAC, given the time constraint of the discussion period. We can try featuring a result in the final version of the paper.
>
> For our UCI regression experiments we added another strong baseline. For the small MLP network we use we were able to compute the full GGN matrix in the Laplace approximation of the main network. We copy the result in the form a figure (https://freeimage.host/i/Jnn6z6g)

---

> ### Author Response · Authors · 2023-11-17
>
> *“Both theoretical and experimental runtimes and memory requirements are not discussed. It is unclear if the proposed models take significantly more time for training or inference, since in theory a forward pass of the hypernetwork is required for every "main" network weight.”*
>
> Regarding the computational time of our method, our methods computational time can be decomposed as follows:
>
> 	Time of our method = hypernetwork training/evaluation (1) + approximate inference (2)
>
> Where \(1\) According to the table below is in practice \~1.2 slower than the vanilla network training. As for \(2\) although approximate inference methods methods are expensive, because in our method they are applied  in the small dimensional INR space in general it takes less time to evaluate.
>
>
> Time experiments (Resnet-18 on cifar100, Batch size=64)
>
> **Our method (main network + INR hypernetwork)**
>
>  Forward 0.0069 ± 0. 0001 (sec)
>
>  Backward 0.0145 ± 0.0084 (sec)
>
>
> **Vanilla Network**
>
>  Forward 0.00463 ± 0.00013 (sec)
>
>  Backward 0.01115 ± 0.00015 (sec)
>
> **Our method (main network + fixed ξ perturbations without evaluating INR)**
>
>  Forward 0.004585 ± 0.00028 (sec)
>
>  Backward 0.00901 ± 0.00127 (sec)
>
>
> As for the overhead in terms of learnable parameters is #W_{inr} (total number of the hypernetwork parameters) #AI_{inr} (number of approximate inference parameters applied on the INR space) which is as we mention in the main paper is in fact much less than #AI_{W} (number of approximate inference parameters applied on the full set of main network weights).
> Performance wise our method is still be competitive w.r.t. methods like ensembles of D networks which at best is D times slower than the vanilla network.
>
> Thus, we believe that our method could be applied to ImageNet.
> Furthermore, because the main overhead of our method is the hypernetwork evaluation we investigated the following alternative training scheme, to further improve our method in terms of time.
> Instead of training the main network weights W and W_INR together we update the W_INR parameters every 10 epochs of the main network training, this significantly reduces the computational overhead of our method and we hypothesize it can scale to IMAGENET models and datasets.
>
> **Training of ResNet18 CIFAR100**
>
>  Full Training Accuracy: 69.01 LL: -2.32 Error: 0.3099 Brier: 0.5181 ECE: 0.222
>
>  Alternative    Accuracy: 68.59 LL: -2.38 Error: 0.3141 Brier: 0.5211 ECE: 0.224
>
>
> By updating the W_INR parameters every 10 epochs of the main network training has some (minor) effect in performance.

---

> ### Author Response · Authors · 2023-11-17
>
> *“Can you further evaluate the effect of INR network size on model performance and quality of the uncertainty estimates? Figure 1 already does this for CIFAR-10 but it would be interesting to see for MLPs on UCI and UCI-gap datasets, with INR-Laplace and INR-RealNVP, and perhaps on wider sets of network sizes. Would also be helpful to include error bars for these figures.”*
>
> We added an ablation w.r.t. INR size following the UCI regression setting in our method. We compare 4 different versions of INR hypernetworks with an increasing number of parameters each,namelly (BIG=2500 MED=625, SMALL=75,XSMALL=10).
>
> (Please see Figure in link: https://freeimage.host/i/JnnsF8G)
>
> From the experiments we can observe that there is a limit to where someone can easily scale the INR hypernetwork and simultaneously gain performance. Individual characteristics play significant role to the INR size (main network size, dataset size, dataset dimension)
>  As for the RealNVP, it was difficult to find a setting where the performance of INR-size can be isolated (different RealNVP sizes,  different training and VI hyperparameters etc)

---

> > ### Author Response · Authors · 2023-11-18
> >
> > Follow up experiment ablation w.r.t. INR size following the UCI regression using SWAG approximate inference
> >
> > (Please see Figure in link: https://freeimage.host/i/uci-swag.JnYqx3v)

---

> > > ### Comment · Reviewer_VEKs · 2023-11-19
> > >
> > > Thank you for your comprehensive response to my initial review.
> > >
> > > > We use linearized Laplace with a GGN approximation to the Hessian. Eq. (4) and (5) correspond to the more general case, which is nevertheless consistent with the rest of the model in theory.
> > >
> > > Thank you for clarifying this. In that case, I think you could perhaps focus the main text on Laplace-GGN and, if you wish, include a derivation with full-Hessian Laplace in the appendix, so as to avoid potential confusion here.
> > >
> > > > We use RealNVP as a way to model [...] $q(\\mathbf{w}\_{\\mathrm{INR}})$. SIREN is the architecture of the hypernetwork, so we can’t really compare the two.
> > >
> > > You are right, I apologize for the mixup here. What I was trying to say is that the "*activation*" in RealNVP corresponds to a series of invertible transformations which I believe would have difficulty recovering a perturbation map with high-frequency changes in magnitude when you compare to the SIREN-based INR-Laplace and INR-SWAG. (More on this later.)
> > >
> > > > We find it very difficult to complete any meaningful set of experiments on KFAC, given the time constraint of the discussion period. We can try featuring a result in the final version of the paper.
> > >
> > > Thank you for taking the time to evaluate Laplace-GGN on the UCI datasets. I am a tiny bit surprised that full GGN is underperforming on some of the UCI-gap datasets given that this was the configuration suggested by Foong et al. In any case, I agree that you should consider including Laplace-KFAC as a baseline in the final version.
> > >
> > > > According to the table below [hyper-network training/evaluation] is in practice ~1.2 slower [...].
> > >
> > > Thank you for these numbers. You also say "because [of the] the small dimensional INR space, in general, it takes less time to evaluate." Would be nice to also provide runtimes for inference. If I understood correctly, a forward pass of the hyper-network is required for every main network weight. Do you batch the weight coordinates when computing perturbations? Do you make a single pass with all coordinates or is this overly expensive?
> > >
> > > > We find that sine/periodic activations [...] slightly outperform hyper-networks with ReLU activations.
> > >
> > > > We compare 4 different versions of INR hypernetworks with an increasing number of parameters [...]
> > >
> > > > In the multiplicative case, $\\nabla\_{\\varepsilon}$ depends on $\\mathrm{w}$, we argue that [...] $\\mathrm{w}$ [...] can pass valuable information to the hyper-network weights.
> > >
> > > I think I might have a clearer picture now with the additional experiments on hyper-network size and ReLU and, more importantly, thanks to your comment on gradient information and multiplicative perturbation.
> > >
> > > Do you think that maybe the main network weights in linear layers re-arrange themselves in each layer to benefit from structural consistency? For example, that weights with low pertubation variance tend to one side of the weight vector and those with high variance to the other? This could be facilitated, as you say, by the multiplicative nature of the perturbation and would also explain why ReLU hyper-networks are also successful.
> > >
> > > I realize that the hyper-network inputs are high dimensional (in particular for the CNN kernels), but an experiment that could demonstrate this would be to observe perturbation variance as a function of the input weight coordinate by taking 1D slices or 2D slices of the input space and seeing if this is highly jagged or is somewhat smooth in nature.

---

> > > > ### Author Response · Authors · 2023-11-21
> > > >
> > > > *“Thank you for clarifying this. In that case, I think you could perhaps focus the main text on Laplace-GGN and, if you wish, include a derivation with full-Hessian Laplace in the appendix, so as to avoid potential confusion here.”*
> > > >
> > > > We understand that introducing the Laplace GGN model could be somehow confusing to the reader at first glance. We will gladly add the full-Hessian Laplace in the appendix.
> > > >
> > > > *“Thank you for taking the time to evaluate Laplace-GGN on the UCI datasets. I am a tiny bit surprised that full GGN is underperforming on some of the UCI-gap datasets given that this was the configuration suggested by Foong et al. In any case, I agree that you should consider including Laplace-KFAC as a baseline in the final version.”*
> > > >
> > > > We will certainly add the KFAC-Laplace for regression in the final version.
> > > >
> > > > *“Thank you for these numbers. You also say "because [of the] the small dimensional INR space, in general, it takes less time to evaluate." Would be nice to also provide runtimes for inference. If I understood correctly, a forward pass of the hyper-network is required for every main network weight. Do you batch the weight coordinates when computing perturbations? Do you make a single pass with all coordinates or is this overly expensive?”*
> > > >
> > > > Yes, a forward pass of the hyper-network is required for every main network, in fact we created an algorithm for both inference and training of our method:
> > > >
> > > > **Algorithm 1** Training
> > > > ```
> > > >  Inputs: I (indices of main network weights), Net (main network), INR (INR hypernetwork), Dataset
> > > >  for number of epochs do
> > > > 	 for x,y in Dataset do
> > > > 		 ξ = INR(I)
> > > > 		 y* = Net(x,ξ)
> > > > 		 loss = (y,y*)
> > > > 		 update INR w.r.t loss
> > > > 		 update Net w.r.t loss
> > > > 	 end
> > > >  end
> > > > ```
> > > >
> > > > **Algorithm 2** Inference
> > > > ```
> > > >  Inputs: I (indices of main network weights), Net (main network), INR (INR hypernetwork), Test_set
> > > >  Ap.-In (Approximate inference method*) MC_samples (Number of Monte Carlo samples)
> > > >  for x in Test_set do
> > > > 	 for MC_samples do
> > > > 		 ξj = Ap.-In(INR, I)
> > > >    	 y* = Net(x,ξj)
> > > > 	 end
> > > > 	 calculate y* statistics
> > > >  end
> > > > ```
> > > >
> > > > (In this setting a post training Monte Carlo-based approximate inference method is implied).
> > > >
> > > > As for the weight coordinates, in practice these values are batched and computed separately for each layer (i.e. for the layer i-th layer indices/input-coordinates positions have the shape [#W, I_dims] where #W is the number of the total main network parameters of the i-th layer and I_dims is the dimensionality of the indices (ex. for convolutional main layer I_dims = 5, 4 positions for the kernel plus 1 dimension to act as a the layerwise position)).
> > > >
> > > > In practice passing all the indices through the hypernetwork at once is more expensive than our approach. Furthermore because the INR is shared between all weights we could run the computations per layer in parallel as they don't depend on each other.
> > > >
> > > > We included runtimes for inference:
> > > >
> > > > Inference time for Resnet18 combined with different stochastic subspaces and different approximate inference methods (time is measured in seconds and for a batch of 10 CIFAR images). We included Inference time comparison for the same approximate inference method but for different ‘subspaces’. Specifically we compare our INR low dimensional space with: Rank-1 (Dusenberry et al. 2020); Wasserstein subnetwork (Daxberger et al. 2021) and partially stochastic ResNets from (Sharma, Mrinank, et al. 2023) we provide you with quantitative results at the the end of this thread.
> > > >
> > > > **For the Laplace approximate inference method**:
> > > >
> > > > Subnetwork: 	0.4211 ± 0.024
> > > >
> > > > INR: 		0.5145 ± 0.008
> > > >
> > > > Rank1: 	0.5545 ± 0.019
> > > >
> > > > Partially: 	0.4994 ± 0.011
> > > >
> > > > All: 		0.4989 ± 0.003
> > > >
> > > > **For the SWAG approximate inference method**:
> > > >
> > > > Subnetwork: 	0.2917 ± 0.0164
> > > >
> > > > INR: 		0.1149 ± 0.0057
> > > >
> > > > Rank1: 	0.2837 ± 0.0027
> > > >
> > > > Partially: 	0.2813 ± 0.0127
> > > >
> > > > All: 		0.3235 ± 0.0315
> > > >
> > > > **( Inference time for ResNet-50 on CIFAR datasets, cf. Figure 4 main paper)**:
> > > >
> > > > Deep Ensembles 	0.9014 ± 0.0273
> > > >
> > > > Dropout		0.0372 ± 0.0066
> > > >
> > > > LL Laplace 		2.0030 ± 0.0073
> > > >
> > > > INR SWAG 		0.6393 ± 0.0184
> > > >
> > > > INR RealNVP 		0.2045 ± 0.0043
> > > >
> > > > For the Deep Ensembles method the obtained values include additional overhead such as ensemble element loading e.t.c. as it is common practice. Furthermore the Linearized LL Laplace is much slower than the other methods as Computing the Jacobian for the ResNet50 reaches the limits of our computational budget at this time.

---

> > > > ### Author Response · Authors · 2023-11-21
> > > >
> > > > *“Do you think that maybe the main network weights in linear layers re-arrange themselves in each layer to benefit from structural consistency? For example, that weights with low pertubation variance tend to one side of the weight vector and those with high variance to the other? This could be facilitated, as you say, by the multiplicative nature of the perturbation and would also explain why ReLU hyper-networks are also successful. I realize that the hyper-network inputs are high dimensional (in particular for the CNN kernels), but an experiment that could demonstrate this would be to observe perturbation variance as a function of the input weight coordinate by taking 1D slices or 2D slices of the input space and seeing if this is highly jagged or is somewhat smooth in nature.”*
> > > >
> > > > We plotted the “perturbation variance” (the values of ξ) as a function of input weight coordinates. Specifically for Resnet18 trained on CIFAR we plotted the flatten values for each specific kernel position across channels (channel slice) for 2 two different convolutional layers.
> > > > Both types of hypernetworks produce well structured perturbation functions.
> > > > The ξ values produced from the sinusoidal hypernetwork are expressed as a somewhat oscillatory behavior w.r.t. channel position, which translates as higher frequency content. As for the ReLU perturbations, while having some high frequencies due to the discontinuity of the ReLU activation, the overall signal has a smooth structure less complicated that the sinusoidal ones in some cases. Unsurprisingly, the ReLU result consists of practically piecewise linear components. This is what we believe that highlights the *marginally* better performance of SIREN hypernetworks.
> > > >
> > > > The plots can be examined here:
> > > >
> > > > https://freeimage.host/i/JnD7Pst
> > > >
> > > > and:
> > > >
> > > > https://freeimage.host/i/JnDYBmQ

---

> > > > ### Author Response · Authors · 2023-11-21
> > > >
> > > > We tried to measure the quality of proposed subspaces in terms of predictive uncertainty. Specifically we compare our INR low dimensional space with: Rank-1 (Dusenberry et al. 2020); Wasserstein subnetwork  (Daxberger et al. 2021) and partially stochastic ResNets from (Sharma, Mrinank, et al. partially).
> > > > We trained (each method) combined with a Resnet18 for 100 epochs in both Cifar10 and Cifar100 datasets while keeping the approximate inference method fixed same across all low dimensional spaces.
> > > >
> > > > * Sharma, Mrinank, et al. "Do Bayesian Neural Networks Need To Be Fully Stochastic?." International Conference on Artificial Intelligence and Statistics. PMLR, 2023.
> > > >
> > > >  **RANK1**
> > > >
> > > >  SWAG - CIFAR10 In-Dist Test data Accuracy: 91.78 LL: -0.4187 Error: 0.082 Brier: 0.1343 ECE: 0.0522
> > > >
> > > >  SWAG - CIFAR10 Corrupted Test data Accuracy: 77.80 LL: -1.2537 Error: 0.222 Brier: 0.3596 ECE: 0.1469
> > > >
> > > >  Laplace - CIFAR10 In-Dist Test data Accuracy: 91.01 LL: -1.5630 Error: 0.090 Brier: 0.6872 ECE: 0.6871
> > > >
> > > >  Laplace - CIFAR10 Corrupted Test data Accuracy: 78.07 LL: -1.7068 Error: 0.220 Brier: 0.7396 ECE: 0.5737
> > > >
> > > > -----------------------------------------------------------------------------------------------------------------------------------------------------
> > > >
> > > > SWAG -   CIFAR100 In-Dist Test data         Accuracy: 65.84 LL: -2.29 Error: 0.34 Brier: 0.55 ECE: 0.228
> > > >
> > > > SWAG -   CIFAR100 Corrupted Test data   Accuracy: 42.80 LL: -4.77 Error: 0.57 Brier: 0.92 ECE: 0.398
> > > >
> > > >  Laplace - CIFAR100 In-Dist Test data        Accuracy: 69.00 LL: -4.01 Error: 0.31 Brier:  0.9714 ECE: 0.668
> > > >
> > > >  Laplace - CIFAR100 Corrupted Test data   Accuracy: 42.00 LL: -4.25 Error: 0.58 Brier 0.979 ECE: 0.401
> > > >
> > > >  **INR**
> > > >
> > > >  SWAG -   CIFAR10 In-Dist Test data Accuracy: 92.17 LL: -0.384 Error: 0.078 Brier: 0.1296 ECE: 0.0498
> > > >
> > > >  SWAG -   CIFAR10 Corrupted Test data Accuracy: 78.60 LL: -1.164 Error: 0.214 Brier: 0.3566 ECE: 0.1447
> > > >
> > > >  Laplace - CIFAR10 In-Dist Test data Accuracy: 89.0 LL: -1.563 Error: 0.110 Brier:  0.6869 ECE: 0.6647
> > > >
> > > >  Laplace - CIFAR10 Corrupted Test data Accuracy: 81.0 LL: -1.660 Error: 0.190 Brier: 0.7156 ECE: 0.5890
> > > >
> > > > ------------------------------------------------------------------------------------------------------------------------------------------------------
> > > >
> > > >  SWAG -   CIFAR100 In-Dist Test data Accuracy: 69.12 LL: -2.094 Error: 0.308 Brier: 0.5006 ECE: 0.2046
> > > >
> > > >  SWAG -   CIFAR100 Corrupted Test data Accuracy: 46.5 LL: -4.1878 Error: 0.535 Brier: 0.8418 ECE: 0.3640
> > > >
> > > >  Laplace - CIFAR100 In-Dist Test data Accuracy: 70.0 LL: -3.9172 Error: 0.300 Brier:  0.967 ECE: 0.6747
> > > >
> > > > Laplace - CIFAR100 Corrupted Test data   Accuracy: 42.0 LL: -4.199 Error: 0.58 Brier:0.9770 ECE: 0.396
> > > >
> > > >  **SUBNETWORK**
> > > >
> > > >  SWAG -   CIFAR10 In-Dist Test data Accuracy: 92.54 LL: -0.4251 Error: 0.074 Brier: 0.1255 ECE: 0.0491
> > > >
> > > >  SWAG -   CIFAR10 Corrupted Test data Accuracy: 76.90 LL: -1.4599 Error: 0.231 Brier: 0.3845 ECE: 0.1732
> > > >
> > > >  Laplace - CIFAR10 In-Dist Test data Accuracy: 91.0 LL: -1.551723 Error: 0.090 Brier: 0.682 ECE: 0.6823
> > > >
> > > >  Laplace - CIFAR10 Corrupted Test data  Accuracy:  81.0 LL: -1.650211 Error: 0.190 Brier: 0.7134 ECE: 0.5886
> > > >
> > > > ------------------------------------------------------------------------------------------------------------------------------------------------------
> > > >
> > > >  SWAG -   CIFAR100 In-Dist Test data Accuracy: 69.86 LL: -2.1430 Error: 0.30 Brier: 0.49 ECE: 0.207
> > > >
> > > >  SWAG -   CIFAR100 Corrupted Test data Accuracy: 49.0 LL: -3.9721 Error: 0.51 Brier: 0.82 ECE: 0.3483
> > > >
> > > >  Laplace - CIFAR100 In-Dist Test data Accuracy: 68.0 LL: -3.9505 Error: 0.32 Brier: 0.9682 ECE: 0.655
> > > >
> > > >  Laplace - CIFAR100 Corrupted Test data Accuracy: 49.0 LL: -4.1309 Error: 0.51 Brier: 0.974 ECE: 0.466
> > > >
> > > >  **PARTIALLY STOCHASTIC**
> > > >
> > > >  SWAG -   CIFAR10 In-Dist Test data Accuracy: 92.54 LL: -0.4251 Error: 0.074 Brier: 0.1255 ECE: 0.0491
> > > >
> > > >  SWAG -   CIFAR10 Corrupted Test data Accuracy: 76.92 LL: -1.4499 Error: 0.201 Brier: 0.3806 ECE: 0.1730
> > > >
> > > >  Laplace - CIFAR10 In-Dist Test data Accuracy: 90.8 LL: -1.561 Error: 0.091 Brier: 0.68 ECE: 0.702
> > > >
> > > >  Laplace - CIFAR10 Corrupted Test data Accuracy: 80.0 LL: -1.67 Error: 0.21 Brier: 0.72 ECE: 0.59
> > > >
> > > >
> > > > ***thread continued.........***

---

> > > > ### Author Response · Authors · 2023-11-21
> > > >
> > > > ----------------------------------------------------------------------------------------------------------------------------------------------------
> > > >
> > > >  SWAG - CIFAR100 In-Dist Test data Accuracy: 69.85 LL: -2.1430 Error: 0.3015 Brier: 0.4997 ECE: 0.2078
> > > >
> > > >  SWAG - CIFAR100 Corrupted Test data Accuracy: 49.0 LL: -3.9721 Error: 0.51 Brier: 0.8218 ECE: 0.3483
> > > >
> > > >  Laplace - CIFAR100 In-Dist Test data Accuracy: 66.0 LL: -3.9903 Error: 0.34 Brier:  0.978 ECE: 0.6377
> > > >
> > > >  Laplace - CIFAR100 Corrupted Test data Accuracy: 49.0 LL: -4.1815 Error: 0.51 Brier 0.989 ECE: 0.4709
> > > >
> > > > \*For SWAG and Linearized Laplace with GGN, in order to be able to run across low dimensional spaces we choose the covariance to have Diagonal structure.
> > > >
> > > > As for the results, we believe that in both datasets there is at the very least a trend in favor of both proposed INR-x methods. The results validate to a considerable degree the premise of the proposed methods: Instead of choosing a subset or subnet following the rationale of the corresponding methods, the INR produces "ξ" outputs that endow the full network with the desirable stochasticity, while keeping the dimensionality of the random process that we want to do inference upon at a low level.

---

> > > > > ### Comment · Reviewer_VEKs · 2023-11-22
> > > > >
> > > > > > [...] INR is shared between all weights, we could run the computations per layer in parallel as they don't depend on each other.
> > > > >
> > > > > This is interesting insight which I think the manuscript would benefit from if made explicit.
> > > > >
> > > > > > We plotted the “perturbation variance” (the values of ξ) as a function of input weight coordinates. Specifically for ResNet-18 trained on CIFAR we plotted the flattened values for each specific kernel position across channels (channel slice) for 2 two different convolutional layers.
> > > > >
> > > > > Thank you for these figures. I was looking for the variance, i.e. $\\mathrm{Var}[\epsilon]$ as a function of input weight coordinates, not $\epsilon$ itself. I was also more interested in a linear layer versus convolution. However, these figures are already quite valuable since they are slicing across the channel axis. If we assume that the variance has similar behavior, this would indicate that kernels with high variance are grouped together to benefit from the smooth nature of the INR function.
> > > > >
> > > > > > We tried to measure the quality of proposed subspaces in terms of predictive uncertainty. [...]
> > > > >
> > > > > I think these results would be of interest to reviewer tKmG. I tend to agree with their assessment that the manuscript could benefit from experiments which differentiate it from prior work on hypernetwork-based methods, and these results seem to be heading in this direction.

---

> ### Author Response · Authors · 2023-11-17
>
> *“Do you have an intuition for why multiplicative perturbations are so successful compared to additive perturbations of the weights? Perhaps uncertainty is improved when perturbations are scaled up by the magnitude of the weight?”*
>
> We believe that the benefits of multiplicative perturbations can be traced to the training procedure of the INR hypernetwork. Specifically consider the following example of a single linear layer with inputs x, outputs y, weights W and perturbations ξ.
>
> For the multiplication  we have y = x*w*ξ and for  additive  y = x*w + x*ξ
> The back-propagated gradients w.r.t. perturbations ξ  ∇ξ (which are responsible for the hypernetwork training) will be ∂L/∂y *x*W and ∂L/∂y*x respectively. Because in the multiplicative structure ∇ξ depends on W, we argue that because W is responsible for fitting the data can pass valuable information to the hypernetwork weights in the multiplicative case leading to significant increase in the over all performance.

---

> ### Author Response · Authors · 2023-11-17
>
> *“Is there a reason for INR-SWAG appearing in some experiments and INR-Laplace in others?”*
>
> In our work we tried to validate our INR-’space’ combined with a variety of approximate inference methods. Both in classification and regression experiments all the three posterior approximations combined our method yield good and calibrated results.
> There are some reasons why some methods do not appear in some figures. For example in Fig1 we wanted to also measure the diversity so we wanted a Monte Carlo based method and to save space we chose SWAG. Also in Fig2 we wanted to highlight the benefits of Laplace approximation in particular. In general we followed a pattern that in Regression we use (Laplace/RealNVP) , and in Classification (SWAG/RealNVP). We will gladly include all three methods in all experiments in the appendix.

---

> ### Author Response · Authors · 2023-11-23
>
> *“This is interesting insight which I think the manuscript would benefit from if made explicit.”*
>
> We agree. We will rework this part of the manuscript to make this point explicit.
>
> *“Thank you for these figures. I was looking for the variance, i.e. as a function of input weight coordinates, not itself. I was also more interested in a linear layer versus convolution. However, these figures are already quite valuable since they are slicing across the channel axis. If we assume that the variance has similar behavior, this would indicate that kernels with high variance are grouped together to benefit from the smooth nature of the INR function.”*
>
> Following the same experimental procedure we plotted alongside the mean values of ξ also their ± variance (as this was computed from the SWAG-diagonal approximate inference method), again as a function of channel coordinates. We can observe that the variance has the same structural properties as the mean values of ξ. Thus we believe it is makes sense for the main network convolutional kernel to take advantage of each structure.
>
> https://freeimage.host/i/JoB5aaa
>
> https://freeimage.host/i/JoB5Ejp

---

> ### Comment · Reviewer_VEKs · 2023-11-23
>
> > Following the same experimental procedure we plotted alongside the mean values of ξ also their ± variance [...]
>
> These new figures are interesting but slightly unexpected. I was expecting to see larger changes in the variance as a function of weight coordinates but it seems to be more or less static. I suppose this might be linked to the multiplicative nature of the perturbations since the resulting main network weight variance then mostly depends on the magnitudes of the main network weights and perturbation means.
>
> I think the manuscript would benefit from an analysis in the dynamics of the weight perturbations maybe including these new figures and performing some more experiments which go in this direction.

---

> > ### Author Response · Authors · 2023-11-23
> >
> > *“These new figures are interesting but slightly unexpected. I was expecting to see larger changes in the variance as a function of weight coordinates ... ”*
> >
> > We decoupled the variance values from the mean and plotted them again as a function of channel weight coordinates. The mean values of ξ have larger fluctuations which in our plot makes the variance look as you said more or less static. By isolating the variance we can observe that it also has meaningful structural properties (Please consider that the variance values were computed using the diagonal-SWAG, a method which has less expressivity in terms of structured uncertainty estimation).
> >
> > https://freeimage.host/i/JoukcF4
> >
> > https://freeimage.host/i/JoukMN9
> >
> >
> > *“I think the manuscript would benefit from an analysis in the dynamics of the weight perturbations maybe including these new figures and performing some more experiments which go in this direction.  ”*
> >
> > Your recommendations for the analysis of perturbation dynamics is excellent for explaining the empirical performance of our method. As per your comments we will certainly extend and add these new figures in the final manuscript.

---

### Official Review · Reviewer_tKmG · 2023-11-10

**Soundness:** 3 good
**Presentation:** 3 good
**Contribution:** 2 fair
**Rating:** 5
**Confidence:** 5

**Summary:**

The authors propose a hypernetwork-based inference mechanism for BNNs, focusing on two key aspects: first, they want the hypernetwork to be compact and reusable across the main network; second, they perform approximate Bayesian inference over the compact hypernetwork. A key part of this work is the fact that the authors also keep around deterministic weights which are modulated by the stochastic nuisances sampled from the hypernetwork, allowing them to maintain good performance.

The method appears to be competitive in experiments.

**Strengths:**

The authors propose a relatively narrow idea: making a small hypernetwork probabilistic and sharing it across the network.

What I like about this paper is the good exploration over ways to do this: the authors both try Laplace approximations, and stochastic weight averaging as pragmatic means to parametrize an approximate posterior over the hypernet. They also propose to use normalizing flows to shape the outputs of this hypernetwork to squeeze a bit more performance out of it.

Empirically, the work shows good performance.

The paper is overall well written and easy to follow and understand.

I also want to call out that the authors are good scholars, the breadth of relevant work cited here is comprehensive and commendable.

**Weaknesses:**

The elephant in the room with this paper is that it is very narrow in its contribution.

Tiny shared hypernernetworks parametrizing individual implicit weight outputs via coordinate systems go as far back as the cited paper Karaletsos et al 2018. Bayesian inference over such hypernetworks has likewise been performed before, in the shape of GPs and BNNs which the authors both cite in their related work.
Blending deterministic and stochastic weights coming from hypernetworks also has been done, again cited absolutely correctly by the authors in numerous papers.

The contribution I see here is not the novelty of any of the ideas then, but the specific engineering combinations tweaked to obtain good performance. For example, executing on practical pieces like Laplace for the hypernetwork or SWA and pairing that object with normalizing flows is good execution that probably helps with performance compared to previously mentioned works.

**Questions:**

Given the relative dearth of new ideas in this work, I would like to identify its strengths in execution and overall quality of the work.

Could the authors argue that their specific combination of techniques could be applied to a larger neural network, i.e. an imagenet model?

Could the authors share a bit more about scalability and performance/memory constraints/ number of forward passes needed to obtain good performance?

I would enjoy seeing more evidence that highlights the merits of the execution here, given the relatively modest technical contributions.

As the paper stands, I would find the contributions somewhat thin, but I would enjoy seeing this line of work with hypernetworks for BNNs be paired with a very strong empirical result in the style I ask for above to enrich the community and hope the authors can find something more to show that differentiates this work more from e.g. Dusenberry et al. empirically, which can also be interpreted as a specific hierarchical weight model (with layer-specific structure).

Again, I commend the authors for their quality scholarship and would enjoy more sound arguments to place this work into the trajectory of papers they have mentioned as a primarily empirical contribution.

---

> ### Author Response · Authors · 2023-11-17
>
> *“Empirically, the work shows good performance. The paper is overall well written and easy to follow and understand. [..] I also want to call out that the authors are good scholars, the breadth of relevant work cited here is comprehensive and commendable. [...] Again, I commend the authors for their quality scholarship and would enjoy more sound arguments to place this work into the trajectory of papers they have mentioned as a primarily empirical contribution.”*
>
> We are happy that you appreciate our effort. Thank you for your comment!

---

> ### Author Response · Authors · 2023-11-17
>
> *“Could the authors argue that their specific combination of techniques could be applied to a larger neural network, i.e. an imagenet model? [..] Could the authors share a bit more about scalability and performance/memory constraints/ number of forward passes needed to obtain good performance?”*
>
> Regarding the computational time of our method, our methods computational time can be decomposed as follows:
>
> 	Time of our method = hypernetwork training/evaluation (1) + approximate inference (2)
>
> Where \(1\) According to the table below is in practice \~1.2 slower than the vanilla network training. As for \(2\) although approximate inference methods methods are expensive, because in our method they are applied  in the small dimensional INR space in general it takes less time to evaluate.
>
>
> Time experiments (Resnet-18 on cifar100, Batch size=64)
>
> **Our method (main network + INR hypernetwork)**
>
>  Forward 0.0069 ± 0. 0001 (sec)
>
>  Backward 0.0145 ± 0.0084 (sec)
>
>
> **Vanilla Network**
>
>  Forward 0.00463 ± 0.00013 (sec)
>
>  Backward 0.01115 ± 0.00015 (sec)
>
> **Our method (main network + fixed ξ perturbations without evaluating INR)**
>
>  Forward 0.004585 ± 0.00028 (sec)
>
>  Backward 0.00901 ± 0.00127 (sec)
>
>
> As for the overhead in terms of learnable parameters is #W_{inr} (total number of the hypernetwork parameters) #AI_{inr} (number of approximate inference parameters applied on the INR space) which is as we mention in the main paper is in fact much less than #AI_{W} (number of approximate inference parameters applied on the full set of main network weights).
> Performance wise our method is still be competitive w.r.t. methods like ensembles of D networks which at best is D times slower than the vanilla network.
>
> Thus, we believe that our method could be applied to ImageNet.
> Furthermore, because the main overhead of our method is the hypernetwork evaluation we investigated the following alternative training scheme, to further improve our method in terms of time.
> Instead of training the main network weights W and W_INR together we update the W_INR parameters every 10 epochs of the main network training, this significantly reduces the computational overhead of our method and we hypothesize it can scale to IMAGENET models and datasets.
>
> **Training of ResNet18 CIFAR100**
>
>  Full Training Accuracy: 69.01 LL: -2.32 Error: 0.3099 Brier: 0.5181 ECE: 0.222
>
>  Alternative    Accuracy: 68.59 LL: -2.38 Error: 0.3141 Brier: 0.5211 ECE: 0.224
>
>
> By updating the W_INR parameters every 10 epochs of the main network training has some (minor) effect in performance.

---

> ### Author Response · Authors · 2023-11-22
> **Additional experiments - ReLU vs Sine on hypernet**
>
> We have completed a set of additional experiments, after discussion with the other reviewers. We copy them to you, as you might appreciate the extra empirical results.
>
> The first set of results is about ablation on the hypernetwork:
>
> (Reviewer VEKs): *“The use of SIREN activation suggests that the hypernetworks need to model high frequency representations of the weight perturbations. The SIREN models presented in [Sitzmann et al., 2020] benefit from consistency and repeating patterns in the signals they are fitting, making interpolation easier. There might be some form of structural consistency in neighboring weight perturbations for CNN kernels, but it seems unlikely for the weights of linear layers. [..] This matter is only very briefly brought up in the discussion of Figures 9 and 10. [..] Have you tried using ReLU hypernetworks? Do these fail to recover high frequency representations of the perturbations compared to SIREN?”*
>
> We trained Resnet18 in both CIFAR10 and CIFAR100 for 100 epochs to evaluate the predictive capabilities of the Sinusoidal hyper-network:
>
>  **CIFAR10**
>
>  RELU_MAP Accuracy: 91.11 LL: -0.4891 Error: 0.088 Brier: 0.1484 ECE: 0.05891
>
>  SINE MAP   Accuracy: 91.70 LL: -0.4449 Error: 0.083 Brier: 0.138 ECE: 0.05401
>
>  **CIFAR100**
>
>  RELU_MAP Accuracy: 67.79 LL: -2.544 Error: 0.3221 Brier: 0.537 ECE: 0.23232
>
>  SINE MAP   Accuracy: 68.49 LL: -2.3990 Error: 0.3151 Brier: 0.527 ECE: 0.2256
>
> We find that Sine/Periodic activations – the “default” choice in SIREN – slightly outperforms a hypernet with ReLU activations. Still, results are very close, though there is a trend in favor of sine in all benchmarks.
> The original motivation behind using the sine activation is related to modeling high-frequency content, which translates as details in structured signals such as images or video [Sitzmann 2020]. We can however see this “in the top of its head”, so to speak: in structured signals we care *more* for low-frequency content, and high-frequency is a “good-to-have” content. We can interpret an input semantically if we see its low frequencies, but not necessarily vice versa. For example, image compression will invariably throw away high frequencies first, and the last frequencies to lose will be the lower ones.
> Our conjecture is as follows: When using an INR to model *perturbations*, we are faced with a different situation, that corresponds to a different “frequency landscape” (perhaps even different than the one of model *weights*). In particular, we do not have any reason to differentiate lower or higher frequency content in any respect. We “care” for all frequencies, so we need to have a good way to model high frequencies as well. Perhaps this is the reason the sine activation gives a small edge over ReLU.

---

> ### Author Response · Authors · 2023-11-22
> **Full GGN in LA for UCI regression**
>
> For our UCI regression experiments we added another strong baseline. For the small MLP network we use we were able to compute the full GGN matrix in the Laplace approximation of the main network. We copy the result in the form a figure (https://freeimage.host/i/Jnn6z6g)

---

> ### Author Response · Authors · 2023-11-22
> **Ablation on INR size**
>
> (Reviewer VEKs): *“Can you further evaluate the effect of INR network size on model performance and quality of the uncertainty estimates? Figure 1 already does this for CIFAR-10 but it would be interesting to see for MLPs on UCI and UCI-gap datasets, with INR-Laplace and INR-RealNVP, and perhaps on wider sets of network sizes. Would also be helpful to include error bars for these figures.”*
>
> We added an ablation w.r.t. INR size following the UCI regression setting in our method. We compare 4 different versions of INR hypernetworks with an increasing number of parameters each,namelly (BIG=2500 MED=625, SMALL=75,XSMALL=10).
>
> (Please see Figure in link: https://freeimage.host/i/JnnsF8G)
>
> From the experiments we can observe that there is a limit to where someone can easily scale the INR hypernetwork and simultaneously gain performance. Individual characteristics play significant role to the INR size (main network size, dataset size, dataset dimension)
>  As for the RealNVP, it was difficult to find a setting where the performance of INR-size can be isolated (different RealNVP sizes,  different training and VI hyperparameters etc)
>
> Follow up experiment ablation w.r.t. INR size following the UCI regression using SWAG approximate inference
>
> (Please see Figure in link: https://freeimage.host/i/uci-swag.JnYqx3v)

---

> ### Author Response · Authors · 2023-11-22
> **Runtime comparison for inference**
>
> *“Thank you for these numbers. You also say "because [of the] the small dimensional INR space, in general, it takes less time to evaluate." Would be nice to also provide runtimes for inference. If I understood correctly, a forward pass of the hyper-network is required for every main network weight. Do you batch the weight coordinates when computing perturbations? Do you make a single pass with all coordinates or is this overly expensive?”*
>
> Yes, a forward pass of the hyper-network is required for every main network, in fact we created an algorithm for both inference and training of our method:
>
> **Algorithm 1** Training
> ```
>  Inputs: I (indices of main network weights), Net (main network), INR (INR hypernetwork), Dataset
>  for number of epochs do
> 	 for x,y in Dataset do
> 		 ξ = INR(I)
> 		 y* = Net(x,ξ)
> 		 loss = (y,y*)
> 		 update INR w.r.t loss
> 		 update Net w.r.t loss
> 	 end
>  end
> ```
>
> **Algorithm 2** Inference
> ```
>  Inputs: I (indices of main network weights), Net (main network), INR (INR hypernetwork), Test_set
>  Ap.-In (Approximate inference method*) MC_samples (Number of Monte Carlo samples)
>  for x in Test_set do
> 	 for MC_samples do
> 		 ξj = Ap.-In(INR, I)
>    	 y* = Net(x,ξj)
> 	 end
> 	 calculate y* statistics
>  end
> ```
>
> (In this setting a post training Monte Carlo-based approximate inference method is implied).
>
> As for the weight coordinates, in practice these values are batched and computed separately for each layer (i.e. for the layer i-th layer indices/input-coordinates positions have the shape [#W, I_dims] where #W is the number of the total main network parameters of the i-th layer and I_dims is the dimensionality of the indices (ex. for convolutional main layer I_dims = 5, 4 positions for the kernel plus 1 dimension to act as a the layerwise position)).
>
> In practice passing all the indices through the hypernetwork at once is more expensive than our approach. Furthermore because the INR is shared between all weights we could run the computations per layer in parallel as they don't depend on each other.
>
> We included runtimes for inference:
>
> Inference time for Resnet18 combined with different stochastic subspaces and different approximate inference methods (time is measured in seconds and for a batch of 10 CIFAR images). We included Inference time comparison for the same approximate inference method but for different ‘subspaces’. Specifically we compare our INR low dimensional space with: Rank-1 (Dusenberry et al. 2020); Wasserstein subnetwork (Daxberger et al. 2021) and partially stochastic ResNets from (Sharma, Mrinank, et al. 2023) we provide you with quantitative results at the the end of this thread.
>
> **For the Laplace approximate inference method**:
>
> Subnetwork: 	0.4211 ± 0.024
>
> INR: 		0.5145 ± 0.008
>
> Rank1: 	0.5545 ± 0.019
>
> Partially: 	0.4994 ± 0.011
>
> All: 		0.4989 ± 0.003
>
> **For the SWAG approximate inference method**:
>
> Subnetwork: 	0.2917 ± 0.0164
>
> INR: 		0.1149 ± 0.0057
>
> Rank1: 	0.2837 ± 0.0027
>
> Partially: 	0.2813 ± 0.0127
>
> All: 		0.3235 ± 0.0315
>
> **( Inference time for ResNet-50 on CIFAR datasets, cf. Figure 4 main paper)**:
>
> Deep Ensembles 	0.9014 ± 0.0273
>
> Dropout		0.0372 ± 0.0066
>
> LL Laplace 		2.0030 ± 0.0073
>
> INR SWAG 		0.6393 ± 0.0184
>
> INR RealNVP 		0.2045 ± 0.0043
>
> For the Deep Ensembles method the obtained values include additional overhead such as ensemble element loading e.t.c. as it is common practice. Furthermore the Linearized LL Laplace is much slower than the other methods as Computing the Jacobian for the ResNet50 reaches the limits of our computational budget at this time.

---

> ### Author Response · Authors · 2023-11-22
>
> (Reviewer VEKs): *“Do you think that maybe the main network weights in linear layers re-arrange themselves in each layer to benefit from structural consistency? For example, that weights with low pertubation variance tend to one side of the weight vector and those with high variance to the other? This could be facilitated, as you say, by the multiplicative nature of the perturbation and would also explain why ReLU hyper-networks are also successful. I realize that the hyper-network inputs are high dimensional (in particular for the CNN kernels), but an experiment that could demonstrate this would be to observe perturbation variance as a function of the input weight coordinate by taking 1D slices or 2D slices of the input space and seeing if this is highly jagged or is somewhat smooth in nature.”*
>
> We plotted the “perturbation variance” (the values of ξ) as a function of input weight coordinates. Specifically for Resnet18 trained on CIFAR we plotted the flatten values for each specific kernel position across channels (channel slice) for 2 two different convolutional layers.
> Both types of hypernetworks produce well structured perturbation functions.
> The ξ values produced from the sinusoidal hypernetwork are expressed as a somewhat oscillatory behavior w.r.t. channel position, which translates as higher frequency content. As for the ReLU perturbations, while having some high frequencies due to the discontinuity of the ReLU activation, the overall signal has a smooth structure less complicated that the sinusoidal ones in some cases. Unsurprisingly, the ReLU result consists of practically piecewise linear components. This is what we believe that highlights the *marginally* better performance of SIREN hypernetworks.
>
> The plots can be examined here:
>
> https://freeimage.host/i/JnD7Pst
>
> and:
>
> https://freeimage.host/i/JnDYBmQ

---

> ### Author Response · Authors · 2023-11-22
>
> We tried to measure the quality of proposed subspaces in terms of predictive uncertainty. Specifically we compare our INR low dimensional space with: Rank-1 (Dusenberry et al. 2020); Wasserstein subnetwork  (Daxberger et al. 2021) and partially stochastic ResNets from (Sharma, Mrinank, et al. partially).
> We trained (each method) combined with a Resnet18 for 100 epochs in both Cifar10 and Cifar100 datasets while keeping the approximate inference method fixed same across all low dimensional spaces.
>
> * Sharma, Mrinank, et al. "Do Bayesian Neural Networks Need To Be Fully Stochastic?." International Conference on Artificial Intelligence and Statistics. PMLR, 2023.
>
>  **RANK1**
>
>  SWAG - CIFAR10 In-Dist Test data Accuracy: 91.78 LL: -0.4187 Error: 0.082 Brier: 0.1343 ECE: 0.0522
>
>  SWAG - CIFAR10 Corrupted Test data Accuracy: 77.80 LL: -1.2537 Error: 0.222 Brier: 0.3596 ECE: 0.1469
>
>  Laplace - CIFAR10 In-Dist Test data Accuracy: 91.01 LL: -1.5630 Error: 0.090 Brier: 0.6872 ECE: 0.6871
>
>  Laplace - CIFAR10 Corrupted Test data Accuracy: 78.07 LL: -1.7068 Error: 0.220 Brier: 0.7396 ECE: 0.5737
>
> -----------------------------------------------------------------------------------------------------------------------------------------------------
>
> SWAG -   CIFAR100 In-Dist Test data         Accuracy: 65.84 LL: -2.29 Error: 0.34 Brier: 0.55 ECE: 0.228
>
> SWAG -   CIFAR100 Corrupted Test data   Accuracy: 42.80 LL: -4.77 Error: 0.57 Brier: 0.92 ECE: 0.398
>
>  Laplace - CIFAR100 In-Dist Test data        Accuracy: 69.00 LL: -4.01 Error: 0.31 Brier:  0.9714 ECE: 0.668
>
>  Laplace - CIFAR100 Corrupted Test data   Accuracy: 42.00 LL: -4.25 Error: 0.58 Brier 0.979 ECE: 0.401
>
>  **INR**
>
>  SWAG -   CIFAR10 In-Dist Test data Accuracy: 92.17 LL: -0.384 Error: 0.078 Brier: 0.1296 ECE: 0.0498
>
>  SWAG -   CIFAR10 Corrupted Test data Accuracy: 78.60 LL: -1.164 Error: 0.214 Brier: 0.3566 ECE: 0.1447
>
>  Laplace - CIFAR10 In-Dist Test data Accuracy: 89.0 LL: -1.563 Error: 0.110 Brier:  0.6869 ECE: 0.6647
>
>  Laplace - CIFAR10 Corrupted Test data Accuracy: 81.0 LL: -1.660 Error: 0.190 Brier: 0.7156 ECE: 0.5890
>
> ------------------------------------------------------------------------------------------------------------------------------------------------------
>
>  SWAG -   CIFAR100 In-Dist Test data Accuracy: 69.12 LL: -2.094 Error: 0.308 Brier: 0.5006 ECE: 0.2046
>
>  SWAG -   CIFAR100 Corrupted Test data Accuracy: 46.5 LL: -4.1878 Error: 0.535 Brier: 0.8418 ECE: 0.3640
>
>  Laplace - CIFAR100 In-Dist Test data Accuracy: 70.0 LL: -3.9172 Error: 0.300 Brier:  0.967 ECE: 0.6747
>
> Laplace - CIFAR100 Corrupted Test data   Accuracy: 42.0 LL: -4.199 Error: 0.58 Brier:0.9770 ECE: 0.396
>
>  **SUBNETWORK**
>
>  SWAG -   CIFAR10 In-Dist Test data Accuracy: 92.54 LL: -0.4251 Error: 0.074 Brier: 0.1255 ECE: 0.0491
>
>  SWAG -   CIFAR10 Corrupted Test data Accuracy: 76.90 LL: -1.4599 Error: 0.231 Brier: 0.3845 ECE: 0.1732
>
>  Laplace - CIFAR10 In-Dist Test data Accuracy: 91.0 LL: -1.551723 Error: 0.090 Brier: 0.682 ECE: 0.6823
>
>  Laplace - CIFAR10 Corrupted Test data  Accuracy:  81.0 LL: -1.650211 Error: 0.190 Brier: 0.7134 ECE: 0.5886
>
> ------------------------------------------------------------------------------------------------------------------------------------------------------
>
>  SWAG -   CIFAR100 In-Dist Test data Accuracy: 69.86 LL: -2.1430 Error: 0.30 Brier: 0.49 ECE: 0.207
>
>  SWAG -   CIFAR100 Corrupted Test data Accuracy: 49.0 LL: -3.9721 Error: 0.51 Brier: 0.82 ECE: 0.3483
>
>  Laplace - CIFAR100 In-Dist Test data Accuracy: 68.0 LL: -3.9505 Error: 0.32 Brier: 0.9682 ECE: 0.655
>
>  Laplace - CIFAR100 Corrupted Test data Accuracy: 49.0 LL: -4.1309 Error: 0.51 Brier: 0.974 ECE: 0.466
>
>  **PARTIALLY STOCHASTIC**
>
>  SWAG -   CIFAR10 In-Dist Test data Accuracy: 92.54 LL: -0.4251 Error: 0.074 Brier: 0.1255 ECE: 0.0491
>
>  SWAG -   CIFAR10 Corrupted Test data Accuracy: 76.92 LL: -1.4499 Error: 0.201 Brier: 0.3806 ECE: 0.1730
>
>  Laplace - CIFAR10 In-Dist Test data Accuracy: 90.8 LL: -1.561 Error: 0.091 Brier: 0.68 ECE: 0.702
>
>  Laplace - CIFAR10 Corrupted Test data Accuracy: 80.0 LL: -1.67 Error: 0.21 Brier: 0.72 ECE: 0.59
>
>
> ***thread continued.........***

---

> ### Author Response · Authors · 2023-11-22
>
> ----------------------------------------------------------------------------------------------------------------------------------------------------
>
>  SWAG - CIFAR100 In-Dist Test data Accuracy: 69.85 LL: -2.1430 Error: 0.3015 Brier: 0.4997 ECE: 0.2078
>
>  SWAG - CIFAR100 Corrupted Test data Accuracy: 49.0 LL: -3.9721 Error: 0.51 Brier: 0.8218 ECE: 0.3483
>
>  Laplace - CIFAR100 In-Dist Test data Accuracy: 66.0 LL: -3.9903 Error: 0.34 Brier:  0.978 ECE: 0.6377
>
>  Laplace - CIFAR100 Corrupted Test data Accuracy: 49.0 LL: -4.1815 Error: 0.51 Brier 0.989 ECE: 0.4709
>
> \*For SWAG and Linearized Laplace with GGN, in order to be able to run across low dimensional spaces we choose the covariance to have Diagonal structure.
>
> As for the results, we believe that in both datasets there is at the very least a trend in favor of both proposed INR-x methods. The results validate to a considerable degree the premise of the proposed methods: Instead of choosing a subset or subnet following the rationale of the corresponding methods, the INR produces "ξ" outputs that endow the full network with the desirable stochasticity, while keeping the dimensionality of the random process that we want to do inference upon at a low level.

---

> > ### Comment · Reviewer_tKmG · 2023-12-04
> > **acknowledging response**
> >
> > Dear authors,
> >
> > I just wanted to let you know I saw and appreciate your response.
> >
> > Thank you,
> > Your reviewer

---

### Meta-Review · Area_Chair_fUEv · 2023-12-09

**Metareview:**

The authors propose a hypernetwork-based method for inference in Bayesian neural networks (BNNs), where the hyper-network is compact and reusable, and the model parameters are factored as a function of deterministic and stochastic components.

I thank the authors and the reviewers for the engaging discussions. Overall, the reviewers found the proposed approach is a interesting novel combination of existing ideas with a relatively good execution. There were some issues raised with regards to clarity, baselines used, execution times and memory requirements and ablations. I believe the authors have been comprehensive in their rebuttal and I think the results and clarifications make this paper worth presenting at ICLR. Hence, I recommend acceptance.

**Justification For Why Not Higher Score:**

Not enough novel methodology.

**Justification For Why Not Lower Score:**

Good combination of ideas with a solid execution.

---

### Decision · Program_Chairs · 2024-01-16

Accept (poster)